# Unequal power outages induced by natural disasters

Bo Wang [1,2], Han Shi[1,2] ✉, Yueming 'Lucy' Qiu [3] ✉, Nana Deng [2,4] ✉, Destenie Nock [5], Xingchi Shen[6], Zhaohua Wang [2,4] ✉ & Yi 'David' Wang [7]

Natural disasters increasingly threaten energy system reliability. However, little empirical research has examined the unequal impact of such events on power outages. Here, we employ a nationwide high-frequency point-level power outage and natural disasters dataset in China, spanning from 2019 to 2021, to empirically assess the impact of natural disasters on power outages. We focus on the poverty counties, identified by the Chinese government, based on income, infrastructure, geographical location, and other criteria. We find that these impacts of natural disasters on power outage are not distributed evenly between poverty counties (5.19% and 8.96% increase in frequency and duration, respectively) and non-poverty counties (3.80% and 5.34%). Long-term projections under SSP-RCP scenarios suggest that climate change exacerbates the disparity. This paper highlights the need for planners to evaluate disaster-induced outages in vulnerable regions to target climate funds to areas with the utmost necessity.

As natural disasters intensify due to climate change and the penetration of non-dispatchable energy generators (e.g., renewables) into the power grid accelerates in response, power outages increase as a result, particularly in developing nations[1-3]. In April 2022, India's electricity demand hit its highest level in 38 years due to high temperatures, forcing power cuts across India and triggering blackouts of up to eight hours a day in some regions[4]. Due to heavy rain and flooding, a power station in South Africa was submerged, leading to a temporary reduction of 6000 megawatts in the national power supply in 2019. During the power restrictions, South African residents faced at least two power outages per day, each lasting at least two hours. It was estimated that the power outages resulted in a daily economic loss of $70.4 million for South Africa[5]. In recent years, there have been a series of unprecedented large-scale power outages due to natural disasters in vulnerable regions of China[6,7]. In May 2019, severe rainfall and flooding in Guangxi's Guilin and Chongzuo regions led to widespread power outages across rural communities. In June 2020, a landslide in Aniangzhai Village, Danba County, Sichuan Province, caused two 10-kilovolt lines to go out, and interrupted electricity supply to 4507 rural households. These disruptions significantly impacted household daily life, underscoring the limited disaster resilience of vulnerable regions compared to urban areas. Empirical evidence linking natural disasters to power outages is significant since natural-disaster-related power outages have wide-ranging consequences, encompassing adverse effects on human health[8], quality of life[9], and labor productivity[10].

Despite the increasing frequency of natural disasters, there is a gap in empirical understanding of the impacts and corresponding implications on electric reliability in developing countries, particularly regarding comparative analyses across disaster types. The existing literature provides substantial insights into the impacts of natural disasters on the technical and infrastructural resilience of power systems[11,12]. Natural disasters such as hurricanes, wildfires, and floods could reduce the electricity supply by destroying the power infrastructure or damaging transmission and distribution lines[13-15]. However, the existing literature lacks systematic comparisons of how different natural disasters affect both the duration and frequency of

[1]School of Management, Beijing Institute of Technology, Beijing, China. [2]Digital Economy and Policy Intelligentization Key Laboratory of Ministry of Industry and Information Technology, Beijing, China. [3]School of Public Policy, University of Maryland College Park, College Park, MA, USA. [4]School of Economics, Beijing Institute of Technology, Beijing, China. [5]Engineering and Public Policy, Carnegie Mellon University, Pittsburgh, PA, USA. [6]School of International and Public Affairs, Shanghai Jiao Tong University, Shanghai, China. [7]Department of Economics, Virginia Polytechnic Institute and State University, Blacksburg, VA, USA. ✉e-mail: 15510009875@163.com; yqiu16@umd.edu; dn57160@126.com; wangzhaohua@bit.edu.cn

power outages. Furthermore, predictive assessments of power outages and their associated economic loss under future climate change scenarios are limited. Addressing these gaps is crucial for developing targeted adaptation strategies and strengthening regional energy resilience.

In addition, the unequal impact of power outages induced by natural disasters and extreme weather remains unclear. Power reliability in rural areas, under-resourced communities, and areas with poor public infrastructure is more susceptible to disruptions caused by natural disasters and extreme weather events[16,17]. Such instability disproportionately affects vulnerable groups, exacerbating existing socioeconomic disparities. For instance, prolonged power outages hinder access to critical medical services, particularly for individuals with chronic illnesses who rely on electrically powered medical devices

or refrigeration for medications[18–20]. Households lacking backup power solutions (e.g., generators or battery storage) cannot preserve perishable goods without refrigeration[21]. Moreover, there is an evident empirical gap in our understanding of the mechanism for this inequality and whether this inequality will widen under plausible scenarios of climate change. The Sustainable Development Goals (SDGs) call for action to ensure access to affordable, reliable, sustainable, and modern energy for all[22]. Therefore, this study focuses on the poverty counties, identified by the Chinese government, based on income, infrastructure, geographical location, and other criteria[23], aiming to provide empirical evidence on the inequitable and heterogeneous impacts of natural disasters on electricity outages, thus highlighting the importance of energy justice. Specifically, this study fills this empirical gap by applying high-frequency point-level power outage

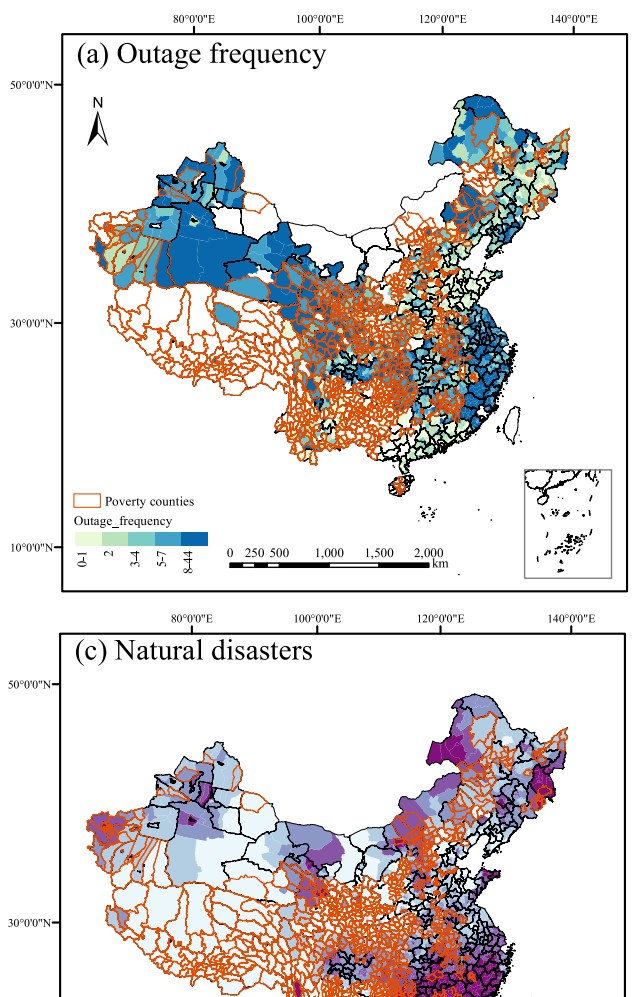

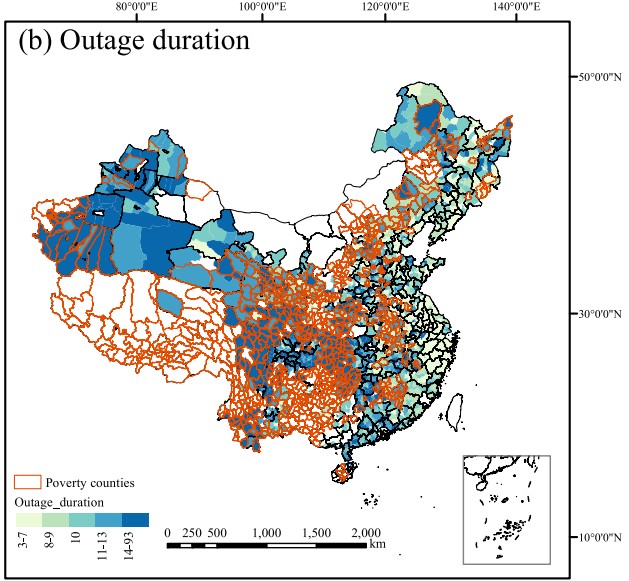

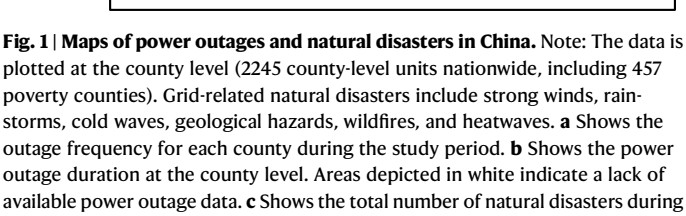

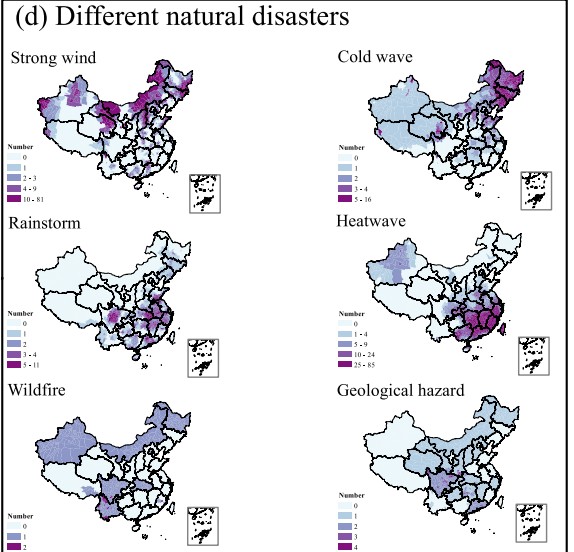

**Fig. 1 | Maps of power outages and natural disasters in China.** Note: The data is plotted at the county level (2245 county-level units nationwide, including 457 poverty counties). Grid-related natural disasters include strong winds, rainstorms, cold waves, geological hazards, wildfires, and heatwaves. **a** Shows the outage frequency for each county during the study period. **b** Shows the power outage duration at the county level. Areas depicted in white indicate a lack of available power outage data. **c** Shows the total number of natural disasters during the study period. **d** Shows different types of natural disasters during the study period. The Orange tracing represents poverty counties. The poverty counties in this study were identified by the Chinese government in 2014. The data span from December 2019 to September 2021. The base map used in this study is the 2019 China map, which has been reviewed and approved with the map review number GS (2019)1822 by the Standard Map Service System of the Ministry of Natural Resources.

data and natural disaster data from December 2019 to September 2021 in China for 2,245 county-level administrative divisions to explore the impact and mechanism of natural disasters on power outages between non-poverty and poverty counties.

In this work, we first compile nationwide high-frequency point-level power outages and a daily county-level severe weather and climate-related natural disasters database in China. The dataset comprises a total of 441,708 outages and 23,600 heatwaves, 4121 cold waves, 4767 strong winds, 3964 rainstorms, 1713 geographical hazards, and 593 wildfires in 2245 counties from December 2019 to September 2021. Next, we provide empirical evidence on how natural disasters and extreme weather events disrupt power reliability. Specifically, we examine the unequal influences of weather and climate-related natural disasters on power outages across poverty versus non-poverty counties. Lastly, we forecast how climate-induced outages may increase under two Shared-Socioeconomic Pathway-Representative Concentration Pathways: SSP-RCP126 and SSP-RCP585. We estimate that from 2022 to 2099, climate change will increase by 0.072 h/person/year under SSP-RCP126 and 0.142 h/person/year under SSP-RCP585 in China, resulting in damages ranging from 192 to 468 billion RMB. Climate change will further widen the disparity in outage hours caused by natural disasters between poverty and non-poverty counties, with an increase of up to 158%. This paper adds to the policy discussion on crafting adaptation strategies to improve the resilience of power infrastructures to address climate change and facilitate sustainable development. In addition, it can also help policymakers form decisions on climate financing allocation while incorporating the unequal impact of power outages from natural disasters.

## Results

### Power outage and natural disaster

Mapping the power outages and natural disasters in China enables us to investigate the spatial distribution of power outage risk and natural disasters. Panels (a) and (b) in Fig. 1 show the maps of power outages in China from December 2019 to September 2021. We use the outage frequency and outage duration as the main indices for analysis. To enable cross-county comparisons and minimize the impact of population differences, the outage frequency for a given day is calculated by aggregating the total number of outages within a county and normalizing it by the county's population. Details of the indices are shown in the Methods section.

As depicted in the bar chart located in the lower-left panel of Panels (a) and (b) in Fig. 1, we find that both the outage frequency and outage duration in officially defined poverty counties are higher than those in non-poverty counties. The poverty counties are identified by the Chinese government, determined through a comprehensive set of assessment factors such as per capita annual income, natural conditions, geographical features, infrastructure, and economic conditions[23]. Poverty counties experience more severe power outages, with a frequency 1.29 times higher and a duration 1.46 times longer than those in non-poverty counties. Our observations also reveal that the outage frequency in poverty counties lasting more than 6 hours is 1.55 times higher than that of non-poverty counties, with a duration 1.34 times longer. Additionally, we find that unplanned power outages occur 1.93 times more frequently in poverty counties than in non-poverty counties, with a duration of 1.29 times longer. Panel (c) in Fig. 1 shows the distribution of grid-related natural disasters during the study period, including strong winds, cold waves, rainstorms, heatwaves, wildfires and geological hazards.

The electricity reliability may be influenced by various factors such as natural disasters, transmission lines, electricity demand, and generating capacity. This study will further estimate the marginal impact of natural disasters on power outage and explore the heterogeneous effects between poverty and non-poverty, using econometric models.

**Table 1 | Impacts of grid-related natural disasters on power outages**

|  | (1) ln(Outage frequency) | (2) ln(Outage duration) | (3) ln(Outage frequency) | (4) ln(Outage duration) |
|---|---|---|---|---|
| Natural_disaster | 0.0404*** | 0.0593*** |  |  |
|  | (0.0029) | (0.0055) |  |  |
| Strong_wind |  |  | 0.0298*** | 0.0438*** |
|  |  |  | (0.0058) | (0.0108) |
| Rainstorm |  |  | 0.0799*** | 0.1163*** |
|  |  |  | (0.0088) | (0.0162) |
| Cold_wave |  |  | 0.0377*** | 0.0663*** |
|  |  |  | (0.0046) | (0.0095) |
| Geo_hazard |  |  | 0.0445*** | 0.0598*** |
|  |  |  | (0.0103) | (0.0196) |
| Wildfire |  |  | 0.0059 | 0.0414 |
|  |  |  | (0.0131) | (0.0330) |
| Heatwave |  |  | 0.0360*** | 0.0499*** |
|  |  |  | (0.0042) | (0.0082) |
| Control | YES | YES | YES | YES |
| County*YM | YES | YES | YES | YES |
| Observation | 1497450 | 1504150 | 1497450 | 1504150 |
| $R^2$ | 0.8410 | 0.2505 | 0.8410 | 0.2505 |

The statistical significance of the coefficients is indicated by asterisks: * denotes p < 0.10, ** denotes p < 0.05, and *** denotes p < 0.01. Standard errors are clustered at the county level and reported below the coefficients. ln(Outage frequency) refers to the natural logarithm of power outage frequency. ln(Outage duration) refers to the natural logarithm of power outage duration. Natural_disaster represents the intensity of natural disasters, which is the sum of Strong_wind, Rainstorm, Cold_wave, Geo_hazard, Wildfire and Heatwave. Strong_wind is a dummy variable that equals 1 when the county experiences strong winds; Rainstorm is a dummy variable that equals 1 when the county experiences rainstorms; Cold_wave is a dummy variable that equals 1 when the county experiences cold waves; Geo_hazard is a dummy variable that equals 1 when the county experiences geological disasters; Wildfire is a dummy variable that equals 1 when the county experiences wildfire; Heatwave is a dummy variable that equals 1 when the county experiences heatwaves. Controls include dummy variables for holidays and weekends. County*YM is the county-month-by-year fixed effect. Observation refers to the sample size. R2 indicates the goodness-of-fit of the regressions. The significance of the regression coefficients is assessed using a two-sided t-test, which evaluates whether the coefficients are statistically different from zero.

### Natural disaster induces more power outages

We explore the impact of natural disasters on power outages using a fixed effects model. Table 1 presents the regression results. Column (1) shows the impact of natural disasters on power outage frequency. Column (2) shows the impact of natural disasters on power outage duration. Column (3) shows the impact of different types of natural disasters on power outage frequency. Column (4) shows the impact of different types of natural disasters on power outage duration. The regression results indicate that, in all model specifications, natural disasters have a statistically significant and positive impact on power outages. Specifically, natural disasters increase outage frequency by 4.04% (Column 1, Table 1) and extend outage duration by 5.93% (Column 2, Table 1) on average for an additional unit increase in natural disasters. All models' specifications include county-month-by-year fixed effect (County*YM), which captures the month-by-year-specific characteristics in each county, such as cultural economic conditions, energy transition policies, and local infrastructure. We cluster the standard errors at the county level. We further explore the impacts of different types of natural disasters on power outages. We find that strong wind, rainstorms, cold waves, geological hazards, and heatwaves significantly increase outage frequency by 2.98%, 7.99%, 3.77%, 4.45%, and 3.60% (Column 3, Table 1), respectively. Additionally, these natural disasters increase in outage duration by 4.38%, 11.63%, 6.63%, 5.98%, and 4.99% (Column 4, Table 1), respectively.

Among these natural disasters, rainstorms have the most significant influence on both the outage frequency and duration (Columns 3 and 4, Table 1). It must be emphasized that the observed impact of rainstorms on power outages could be a result of the combined effects of rainstorms and lightning due to the collinearity issues between these two factors. Rainstorms and lightning strike electrical equipment and disrupt the power supply network, thereby exacerbating the vulnerability to power outages. Notably, we find no substantial impact from wildfires. This may be due to the relatively infrequent occurrence and moderate severity of wildfires in China. Based on the wildfire data in 2020, the United States experienced 51 times more forest fires than China, with an economic loss 161 times higher[24,25]. To further eliminate the influence stemming from other confounding factors, we conduct a reanalysis controlling for, and thus excluding the central environmental protection inspections (CEPI), coal price fluctuations, and the impact of COVID-19 on power outages (Supplementary Table 5). The reanalysis results are consistent with our main results, indicating that all grid-related natural disasters, except wildfires, result in increased outages. Given that power outage data were not collected for some counties, we employed the Heckman two-step estimation method to test the robustness of our results. The results are consistent with our main results, indicating their robustness (Supplementary Table 6).

We further compare the impact of natural disasters on planned and unplanned outages. Planned outages are pre-scheduled and notified in advance, while unplanned outages are unexpected and can be triggered by both natural and human-induced factors. We find that natural disasters lead to a 3.50% increase in the frequency of unplanned outages and a 1.04% increase in planned outages (Supplementary Table 7). Natural disasters extend the duration of unplanned outages by 5.78% and planned outages by 1.65% (Supplementary Table 8). Although natural disasters also contribute to an increase in planned outages, this is likely due to heightened maintenance efforts aimed at stabilizing the power grid and mitigating potential risks. The T-test results comparing the coefficients of the two types of outages reveal a significant difference, indicating that unplanned outages are far more strongly impacted by natural disasters than planned outages (Supplementary Table 9).

Additionally, we compare the impact of natural disasters on power outages in counties with varying frequencies of natural disasters. We categorize the cumulative number of natural disasters during the study period into two groups: 0-10 occurrences, and more than 10 occurrences during the study period. The results show that counties with more frequent natural disasters experience a greater impact on power outages due to these disasters (Supplementary Table 10). Specifically, natural disasters increase outage frequency by 1.65% and extend outage duration by 3.13% on average for an additional unit increase in natural disasters during the regions experiencing 0-10 natural disasters in our study period (Supplementary Tables 7 and 8). Natural disasters increase outage frequency by 4.43% and extend outage duration by 6.35% on average for an additional unit increase in natural disasters during the regions of more than 10 natural disasters in our study period.

We also explore the impact of power outages caused by natural disasters across different durations of outages, regions and seasons (Supplementary Tables 7, 8, 9 and 10). Natural disasters increase outage frequency by 2.20% for outages lasting less than 6 hours and by 2.54% for those exceeding 6 hours. Additionally, natural disasters extend outage duration by 2.51% for outages under 6 hours and by 5.20% for outages over 6 hours. Natural disasters in southern China lead to a substantial 5.52% increase in outage frequency and an 8.32% extension in outage duration. In contrast, northern China shows a more modest rise, with outage frequency increasing by 1.08% and duration extending by 1.22%. Southern China experiences a higher overall occurrence rate of natural disasters compared to the north,

particularly with events such as heatwaves and rainstorms, which exert substantial pressure on the power grid system. Natural disasters in summer lead to a 4.52% increase in outage frequency, along with a 6.26% extension in duration. In contrast, during winter, there is a 1.91% escalation in outage frequency and a 3.76% increase in duration.

## Power outages caused by natural disasters in poverty counties

The estimates of the disproportionate impact of natural disasters on the outage frequency and duration in both poverty and non-poverty counties are presented in Panels (a) and (b) of Fig. 2. The poverty counties exhibit a higher frequency and longer duration of power outages caused by natural disasters compared to non-poverty counties. Specifically, natural disasters increase the outage frequency by 5.19% for poverty counties, and by 3.80% for non-poverty counties. Natural disasters extend the outage duration, resulting in an average increase of 8.96% for poverty counties and 5.34% for non-poverty counties. We also examine the unequal power outages caused by natural disasters by introducing interaction terms of natural disasters and the dummy variable of poverty counties(Supplementary Table 11). Additionally, we use the alternative definition of poverty counties and natural disasters to show the robustness of the results (Supplementary Table 11). Considering the different baseline levels of outages in different counties, we examine the impact of natural disasters on incremental values of power outages (Supplementary Tables 2 and 3). The results show that poverty counties experience more outage frequency and longer outage duration caused by natural disasters, showing the robustness of the results. We also construct a more comprehensive indicator (Total_outage_hours), which simultaneously captures the frequency and duration characteristics of power outages (see Methods section). The results are highly consistent with baseline results (Supplementary Table 4). Considering the importance of outage loss, we further compare the marginal economic losses caused by power outages due to natural disasters in poverty and non-poverty counties by the average proportion of outages impacting various loads in poverty and non-poverty counties and the corresponding economic losses for each load. We find that the marginal losses in poverty counties are 1.56 times higher than those in non-poverty counties, based on the estimated parameters of natural disasters on Total_outage_hours (Supplementary Table 4).

Plausible explanations for the lower electricity reliability in responding to natural disasters for the poverty counties may be the lower level of development of the local grid infrastructure[26] and the maintenance capacity[27] of the grid, both of which likely play a key role in ensuring power stability. We use electric grid investment data as a proxy variable for assessing the development of grid infrastructure, while utilizing the labor force of power supply stations at the county level as a proxy variable to evaluate the maintenance capability of the grid network to validate the importance of both mechanisms to power outages. Our results (Supplementary Table 12) indicate that increased investment in grid infrastructure and the augmentation of the workforce at county power supply stations can effectively mitigate the frequency and duration of power outages during natural disasters. Given the vulnerable power grid infrastructure construction and limited power maintenance capacity in poverty areas (the labor force in county power supply stations for poverty counties is 0.38 times that in non-poverty counties), the occurrence of natural disasters leads to higher frequency and longer duration of power outages in these regions.

We further explore the impacts of different types of disasters on power outages in poverty and non-poverty counties. Panels (a) and (b) of Fig. 2 show the impact of different natural disasters on outage frequency and outage duration. Specifically, in poverty counties, the strong wind, rainstorms, cold waves, geological hazards, and heatwaves increase outage frequency by 6.67%, 14.48%, 6.27%, 6.14%, and 2.96%, respectively, compared to 2.26%, 7.07%, 3.27%, 3.90% and

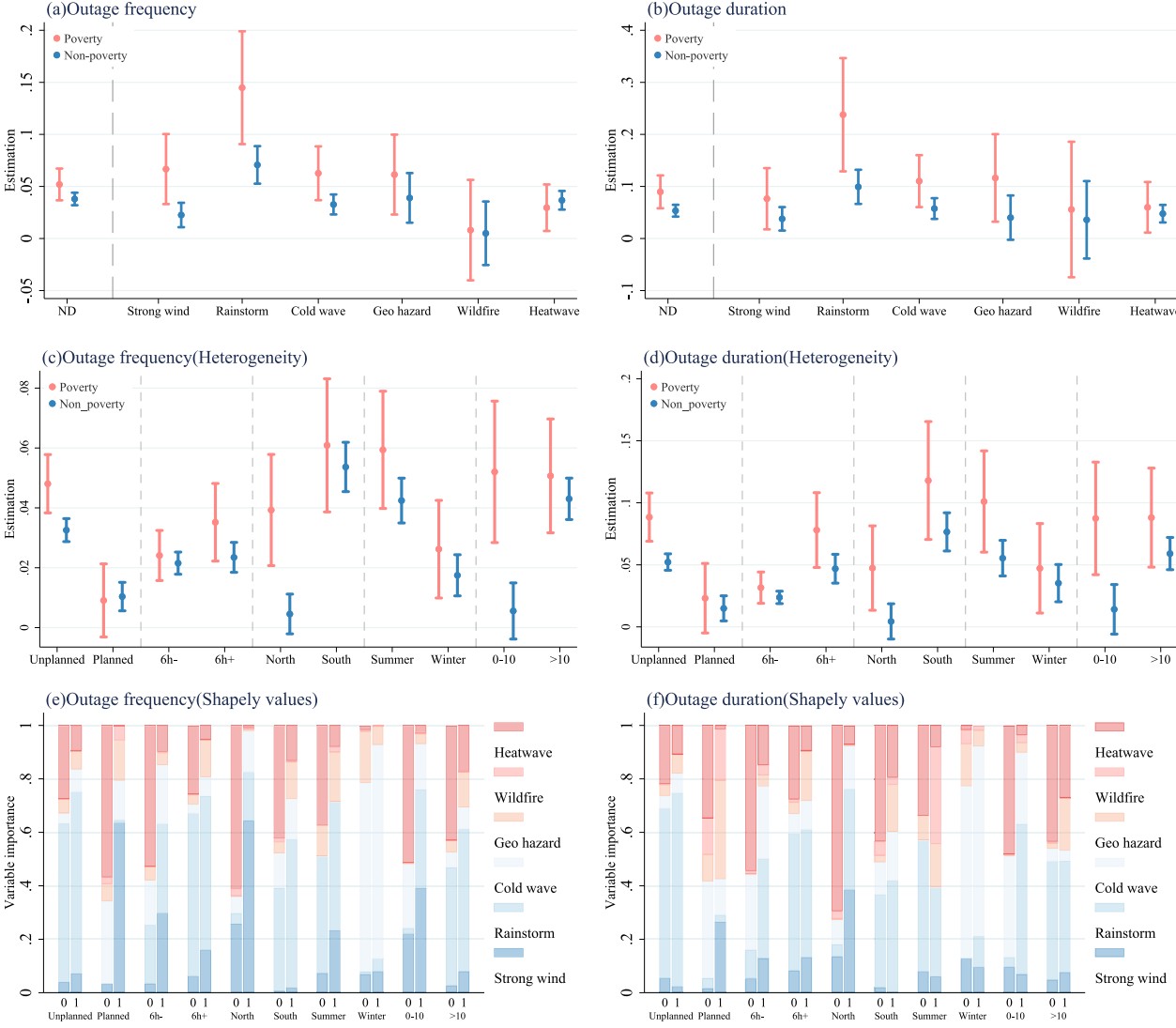

**Fig. 2 | Heterogeneity of power outages caused by natural disasters.** Notes: In **a** and **b**, we show the impacts of Natural disaster, Strong wind, Rainstorm, Cold wave, Geo hazard, Wildfire and Heatwave on power outages. In **c** and **d**, we show the heterogeneity impact of Natural disaster on power outages. The solid dots denote the point estimation coefficients and the lines refer to the 95% confidence interval of the coefficients in **a–d**. In **e** and **f**, we show the decomposition results of natural disaster impacts on power outages based on Shapley Values. The definition of summer is from May to October. The definition of winter is the remaining months. 0-10 represents counties with fewer than 10 natural disaster occurrences, whereas >10 indicates counties experiencing more than 10 natural disasters during the study period. The 1 and 0 in the X-axis in **e** and **f** represent the poverty and non-poverty counties, respectively. Our regression analysis dataset comprises data from 2245 counties across China. The significance of the regression coefficients is assessed using a two-sided t-test, which evaluates whether the coefficients are statistically different from zero. Source data can be found in sheets Supplementary Data 2–7 of file Supplementary Data 1–7.

3.67% in non-poverty counties. Furthermore, these natural disasters extend outage duration by 7.64%, 23.78%, 11.02%, 11.64%, and 6.00% in poverty counties, while in non-poverty counties the increases are 3.78%, 9.92%, 5.75%, 4.01% and 4.77%. The Empirical P-Values indicate that poverty counties are more vulnerable to power outages caused by strong winds, rainstorms, and cold waves (Supplementary Table 13).

To explore potential variations across subgroups, we also conduct heterogeneity analyses. Panels (c) and (d) in Fig. 2 show the results. Empirical P-Values indicate that poverty counties experience a higher frequency and longer duration of power outages caused by natural disasters compared to non-poverty counties, particularly in several specific scenarios: unplanned outages, outages lasting more than 6 hours, outages in northern China, outages during the summer season, and in areas with 0 to 10 outage occurrences during the study period (Supplementary Table 13).

Based on the Shapley value approach, we find that 43.86% of the impact of natural disasters on power outage frequency is attributed to rainstorms, and 31.63% is attributed to heatwaves. Similarly, 42.86% of the impact of natural disasters on power outage duration is attributed to rainstorms, while 28.05% is attributed to heatwaves, as indicated by the Shapley value. The primary factor contributing to the outage frequency caused by natural disasters in poverty-stricken counties is rainstorms, accounting for 50.49%, followed by strong winds at 18.00%. The primary factor contributing to power outage duration caused by natural disasters is rainstorms, accounting for 49.61%, followed by cold waves at 15.07% in poverty counties. In non-poverty counties, rainstorm is the primary contributor influencing power outage frequency caused by natural disasters at 41.71%, followed by heatwaves at 38.94%. In terms of power outage duration caused by natural disasters, rainstorms have the most significant contribution at 41.53%, followed by heatwaves at 33.07% for non-poverty counties. We

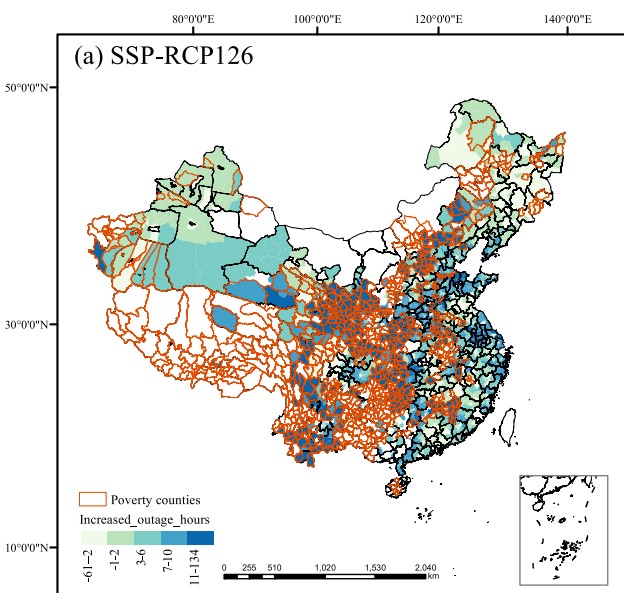
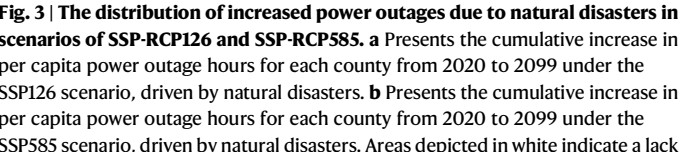
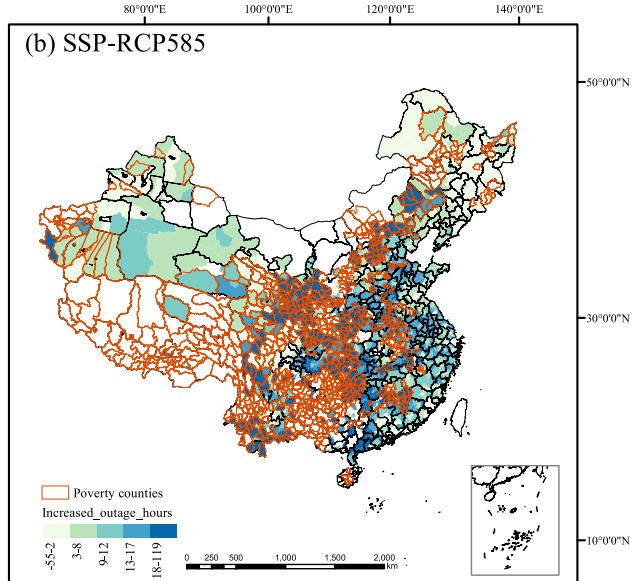

**Fig. 3 | The distribution of increased power outages due to natural disasters in scenarios of SSP-RCP126 and SSP-RCP585. a** Presents the cumulative increase in per capita power outage hours for each county from 2020 to 2099 under the SSP126 scenario, driven by natural disasters. **b** Presents the cumulative increase in per capita power outage hours for each county from 2020 to 2099 under the SSP585 scenario, driven by natural disasters. Areas depicted in white indicate a lack of available power outage data. The color indicates the intensity of the power outages. The orange tracing represents poverty counties. The base map used in this study is the 2019 China map, which has been reviewed and approved with the map review number GS (2019)1822 by the Standard Map Service System of the Ministry of Natural Resources.

also provide the contribution of rainstorms alone, strong winds alone, and their simultaneous occurrence to power outages (Supplementary Fig. 1).

Panels (e) and (f) in Fig. 2 illustrate the heterogeneous results of contribution decomposition based on Shapley values. Rainstorms significantly contribute to unplanned outages both in poverty and non-poverty counties. Cold waves and heatwaves have a greater impact on planned outages in non-poverty counties, while strong winds and geological hazards have a greater contribution to poverty counties. Rainstorms make a significant contribution to more-than-6-hour power outages. In northern regions, strong winds exacerbate outages in poverty counties, while heatwaves intensify outages in non-poverty counties. Winter power outages are primarily driven by cold waves. Heatwaves are the second-largest contributor to summer power outages in non-poverty counties, but their contribution to summer power outages in poverty counties is significantly smaller. Supplementary Fig. 1 also provides the contributions of more detailed types of natural disasters to the impact on power outages.

**Projections under different scenarios**

In this section, we predict the power outage hours caused by natural disasters under different climate scenarios. We obtain the data from the Coupled Model Intercomparison Project 6 (CMIP6). We use the daily meteorological projections under two Shared-Socioeconomic Pathway-Representative Concentration Pathways (SSP-RCP): SSP-RCP126 and SSP-RCP585. The SSP-RCP126 scenario represents a low-emission trajectory, while the SSP-RCP585 scenario depicts a high greenhouse gas emission pathway. We calculate the change of future natural disasters from 2022-2099 compared to the current period (2019-2021) based on the scenarios' data, including maximum wind speed, maximum temperature, minimum temperature, average temperature and precipitation.

We focus on the change in power outage hours caused by strong winds, heatwaves, cold waves, and rainstorms. We quantify the impact of natural disasters on power outage hours between poverty and non-poverty counties based on Eq. (12). The change in outage hours

(compared to the period 2019-2021) is determined by multiplying the change of each type of natural disaster with its respective natural disaster-induced outage hours coefficients, as derived from our previous findings presented in Supplementary Table 4. The calculation assumes that the grid infrastructure remains unchanged from its current state. Considering the population affected by each power outage, the distribution of the increased outage hours caused by natural disasters in scenarios of SSP-RCP126 and SSP-RCP585 from 2022-2099 is shown in Fig. 3.

The projected increase in power outages is expected to result in 0.072 h/person/year from 2022 to 2099 under the SSP-RCP126 scenario, and 0.142 h/person/year under the SSP-RCP585 scenario. We also perform a back-of-the-envelope calculation of the economic cost of power outages induced by natural disasters (see Methods). Based on our calculations, the projected damages from 2022 to 2099 are estimated to range from 192 to 468 billion RMB. And we find that climate change will further widen the outage gaps caused by natural disasters between poverty and non-poverty counties in SSP-RCP126 (157.82%) and SSP-RCP585 scenarios(157.76%). These findings indicate that the reliability of power supply in poverty counties will face substantial challenges in the future.

Building upon the calculation of economic costs using the original population-based method, we also incorporate a total GDP-based analysis as a comparative benchmark. The new projected damages between 2022 and 2099 are estimated to range from 257 to 481 billion RMB, which is slightly higher than the losses of 192 to 468 billion RMB calculated considering the population affected by power outages.

In the Supplementary section titled "Projections under different scenarios", we explore the potential impact of improvements in emergency response capabilities and grid infrastructure upgrades on power outages caused by natural disasters. Apart from the Reference Scenario, we also construct two scenarios: the Improvement to the Non-Poverty Counties Level Scenario and the Improvement to the Top 20% Level Scenario. The results indicate that the improvement of emergency response capabilities and grid infrastructure significantly reduces the increase in outage hours caused by the changes in natural

disasters compared to the Reference Scenarios. Furthermore, the gaps between poverty and non-poverty counties in the Improvement to Non-Poverty Counties Level Scenario and the Improvement to the Top 20% Level Scenario also reduce compared to the Reference Scenarios (see Supplementary Table 14). This estimation provides us with an indication of the potential damage that can be mitigated through investments in enhanced power systems.

## Discussion

Developing nations disproportionately bear the burden of more frequent and more ferocious extreme weather events, which have resulted in significant electricity shortages and power outages. Accurately assessing the impacts of natural disasters on power outages in these regions is crucial for the timely and efficient allocation of climate financing, such as loss-and-damage funding, to areas with the utmost necessity, thereby bolstering climate mitigation and adaptation capacities.

This study uses nationwide county and daily level natural disasters and power outage data in China to provide empirical evidence on the impact of natural disasters on electric power reliability. Our findings demonstrate that natural disasters significantly amplify both the frequency and duration of power outages. Natural disasters increase the outage frequency by 4.04% while extending the outage duration by 5.93% on average as additional units of natural disasters increase. The impacts of natural disasters on power outages are unevenly distributed, predominantly affecting poverty counties, where there is a 5.19% increase in outage frequency and an 8.96% increase in outage duration. Conversely, for non-poverty counties, the impact is less severe, with a 3.80% increase in outage frequency and a 5.34% increase in outage duration. In the long term, under SSP-RCP scenarios, climate change is projected to exacerbate the disparity in outage impacts between poverty and non-poverty counties, with potential gaps widening by up to 158%. This paper contributes to the discussion on the potential challenges that future climate change may pose to the costs associated with power outages and regional disparities. These empirical findings could help designers mitigate the adverse impact of natural disasters on electricity reliability, especially in poverty-stricken countries by enhancing the resilience of energy infrastructure.

Frequent and intense natural disasters disrupt electricity stability in many countries[2,28]. Our findings are relevant not only to China but also to any region seeking solutions to enhance the reliability of electricity supply with frequent and intense natural disasters. Given the escalating frequency of natural disasters and extreme weather events, coupled with the growing integration of renewable energy sources in the power generation portfolio, future challenges are anticipated to arise from power outages. Our research underscores that natural disasters can profoundly affect the reliability of power systems, particularly in poverty-stricken countries, where these areas often overlap with regions inhabited by low-income populations in developing countries. This can lead to potential economic and health losses, exacerbating the challenges of global warming. Policymakers need to assess the financial repercussions of power outages resulting from natural disasters, identify high-risk areas, analyze the patterns and timing of outages, and implement effective policies to enhance electricity reliability against natural disasters.

The government should actively engage in a range of measures and policy strategies to effectively mitigate the disproportionately amplified natural-disaster-induced power outages in poverty counties. First, upgrading infrastructure should be prioritized, particularly for areas that are susceptible to frequent power outages and natural disasters. Specifically, regions with outdated or inadequate power grids, which are often found in poverty areas, should be the focus of comprehensive construction and maintenance initiatives. Second, the implementation of more sophisticated and advanced inspection strategies, such as leveraging predictive analyses, is crucial. These

strategies, along with effective post-disaster recovery and reconstruction plans, will expedite the restoration of power in affected areas. This process may require collaboration among governments, non-governmental organizations (NGOs), commercial enterprises, and other stakeholders. Third, when planning and designing the power system, it is imperative to consider the potential impact of natural disasters on electricity reliability. Systems should be built to withstand such disasters as robustly as possible, with a particular focus on enhancing the reliability of power grids in poverty-stricken countries. The equity goals should serve as a crucial factor in formulating strategies to prioritize grid upgrades.

## Methods
### Data

The natural disaster data used in this study were obtained from the National Cryosphere Desert Data Center (NCDC) and the Meteorological Data Service Centre of China (MDSCC). In this study, we focused specifically on grid-related natural disasters, including wildfires, geological hazards, heatwaves, cold waves, strong winds, and rainstorms. They can destroy the power infrastructure, damage transmission and distribution lines, and cause a mismatch between power supply and demand, thus causing power outages[29,30]. We identified the wildfires and geological hazards at the daily county level from the forewarning information of NCDDC. And the heatwaves, cold waves, strong winds, and rainstorms were calculated using the weather data from 296 meteorological stations across China in MDSCC. The original weather data from 296 meteorological stations includes average temperature, minimum temperature, maximum temperature, maximum wind speed, and precipitation, as well as the longitude and latitude information of the stations. We used the Inverse Distance Weighting method to match the weather data with the 2245 counties, which took the weighted average of weather data from 296 stations and matched them with the 2245 counties. The statistical description of variables can be found in Supplementary Table 1.

The high-resolution power outage data in China were collected from local public utility websites. Given the significant impact of power outages on both residential, industrial, and commercial activities, the power outage data is made publicly available (though they are not being systematically collected and stored in a central database), and the users can get the real-time power outage information from the local public utility websites. We aggregated power outage data at the county and daily levels, covering the period from December 2019 to September 2021. The outage data includes the location of the power outage, starting time, ending time, and type of power outage (planned or unplanned). Here, planned outages are notified in advance. Unplanned power outages are unexpected power failures, which may be caused by natural and man-made factors[31,32].

In this study, our focus lies on the 457 poverty counties designated by the Chinese government in 2014, known as national poverty counties[23]. The primary determinant for delineating poverty areas is their per capita annual income as well as natural conditions, geographical features, infrastructure and economic conditions[23]. We obtained the future meteorological data from the Community Climate System Model (CCSM), supported by the Beijing Climate Center Climate System. Based on the BCC-CSM2-MR model, we used the daily meteorological projections under two Shared-Socioeconomic Pathway-Representative Concentration Pathways: SSP-RCP126 and SSP-RCP585.

There are many outage records across different locations within a county for a given date. Since the population varies significantly across counties, the number of outage records may be correlated with population size. To address this, we calculate the daily outage frequency by aggregating the total number of outages within a county and normalizing it by the county's population. The outage duration for a given day is calculated by averaging the hours of all outage records

within the county. In addition, we construct an indicator (Total_outage_hours) to simultaneously capture both outage frequency and duration characteristics. This indicator equals the per capita outage frequency (Outage_frequency) in the county on that day multiplied by the average duration per outage (Outage_duration). As an example, we use the power outage situation in Pianguan County, Shanxi, on October 15th, 2021, to illustrate the calculation steps for Outage frequency, Outage duration and Total_outage_hours. Pianguan County experienced two power outages on that day: the first one lasting 2.75 hours, and the second one lasting 7.5 hours. The total population of Pianguan County is 7.3382 (in ten thousand units). The Outage_frequency is calculated to be 0.27 (2/7.3382). And the Outage_duration is calculated to be 5.125 ((2.75 + 7.5)/2) hours. The Total_outage_hours equals to 1.40 (2/7.3382*(2.75 + 7.5)/2).

## Base regression model

This study employs a fixed effects model to investigate the impact of natural disasters on power outages. Natural disasters can be seen as exogenous to county-level administrative divisions. We compare the power outages during and not during natural disasters by controlling for the county-month-by-year fixed effect. Equations (1)–(4) shows our basic regression model.

$$\ln\left(\text{Outage frequency}_{id}\right) = \alpha_1 \text{Natural\_disaster}_{id} + \mathbf{X}'\theta + \delta_{iym} + \varepsilon_{id} \quad (1)$$

$$\ln\left(\text{Outage frequency}_{id}\right) = \beta_1 \text{Strong\_wind}_{id} + \beta_2 \text{Rainstorm}_{id} \\ + \beta_3 \text{Cold\_wave}_{id} + \beta_4 \text{Geo\_hazard}_{id} \\ + \beta_5 \text{Wildfire}_{id} + \beta_6 \text{Heatwave}_{id} + \mathbf{X}'\theta + \delta_{iym} + \varepsilon_{id} \quad (2)$$

$$\ln\left(\text{Outage duration}_{id}\right) = \alpha_1 \text{Natural\_disaster}_{id} + \mathbf{X}'\theta + \delta_{iym} + \varepsilon_{id} \quad (3)$$

$$\ln\left(\text{Outage duration}_{id}\right) = \beta_1 \text{Strong\_wind}_{id} + \beta_2 \text{Rainstorm}_{id} \\ + \beta_3 \text{Cold\_wave}_{id} + \beta_4 \text{Geo\_hazard}_{id} \\ + \beta_5 \text{Wildfire}_{id} + \beta_6 \text{Heatwave}_{id} + \mathbf{X}'\theta + \delta_{iym} + \varepsilon_{id} \quad (4)$$

$$\text{Natural\_disaster}_{id} = \text{Strong\_wind}_{id} + \text{Rainstorm}_{id} + \text{Cold\_wave}_{id} \\ + \text{Geo\_hazard}_{id} + \text{Wildfire}_{id} + \text{Heatwave}_{id} \quad (5)$$

Where $\ln(\text{Outage frequency}_{id})$ denotes the natural logarithm of power outage frequency on day $d$ in county $i$. $\ln\left(\text{Outage duration}_{id}\right)$ denotes the natural logarithm of power outage duration on day $d$ in county $i$. The study includes all types of power outages (such as planned and unplanned power outages) and investigates the overall effects while also differentiating them in separate analyses. Natural disaster$_{id}$ represents the intensity of natural disasters, which is the sum of Strong_wind$_{id}$, Rainstorm$_{id}$, Cold_wave$_{id}$, Geo_hazard$_{id}$, Wildfire$_{id}$ and Heatwave$_{id}$. Strong_wind$_{id}$ (Rainstorm$_{id}$, Cold_wave$_{id}$, Geo_hazard$_{id}$, Wildfire$_{id}$ and Heatwave$_{id}$) is a dummy variable indicating whether a county experiences strong winds (rainstorms, cold waves, geological disasters, wildfire and heatwaves) on day $d$ in county $i$ and 0 otherwise. The identification of wildfires and geological hazards is based on the fore-warning information of NCDDC. Heatwaves are defined as three consecutive days with a maximum temperature of 35 °C or above. Heatwaves have been shown to impact energy systems through increased demand and reduced supply capacity[29,33]. Cold waves are defined as a day's low temperature of less than 4 °C and a temperature decrease of more than 8 °C compared to the previous day. The cold waves have a significant impact on electricity demand, generator sets, and grid infrastructure[34]. Strong winds are defined as a day's maximum wind speeds above 10.8 m/s, which will tremble the power wires and have the potential risk of inducing a power outage[35]. Rainstorms are defined as precipitation of more than 50 mm in a single day. Rainstorms,

especially accompanied by thunder and lightning, can have a serious impact on grid infrastructure[30]. The coefficient $\alpha_1$ measures the marginal impact of natural disasters on power outages. The covariates $\mathbf{X}$ control the holiday and weekend dummy variables. The county-month-by-year fixed effect ($\delta_{iym}$) controls for month-by-year-specific shocks in each county, such as cultural economic conditions, energy transition policies and local infrastructure. We cluster the standard errors at the county level, which is consistent with the sampling of our data.

## The heterogeneity effect caused by natural disasters

The study also examines the unequal power outages caused by natural disasters by introducing interaction terms of natural disasters and the dummy variable of poverty counties. We estimate the heterogeneity effect caused by natural disasters with the following model:

$$\ln\left(\text{Outage frequency}_{id}\right) = \alpha_1 \text{Natural\_disaster}_{id} \\ + \alpha_2 \text{Natural\_disaster}_{id}*\text{Poverty}_i \\ + \mathbf{X}'\theta + \delta_{iym} + \varepsilon_{id} \quad (6)$$

$$\ln\left(\text{Outage duration}_{id}\right) = \alpha_1 \text{Natural\_disaster}_{id} \\ + \alpha_2 \text{Natural\_disaster}_{id}*\text{Poverty}_i \\ + \mathbf{X}'\theta + \delta_{iym} + \varepsilon_{id} \quad (7)$$

Where $\alpha_2$ represents the additional impacts of natural disasters on power outage in poverty counties, that is, the incremental changes in power outage frequency or duration in poverty counties compared to non-poverty counties when natural disasters occur. Other variables share the same definition as in Eqs. (1)–(4). The standard errors are clustered at the county level.

## Decomposition of natural disaster impacts

This study uses the Shapley value decomposition to explore the disparities in the impact of different natural disasters on power outages. The Shapley value decomposition is a cooperative game-based approach for quantifying the contribution of multiple explanatory factors to the dependent variable based on the regression model. The Shapley value decomposition accounts for the effect of variable order on $R^2$ and can accurately assess the contribution of explanatory factors. Compared to standardizing all variables of different types of disasters and including them all in the model, the Shapley value will not change the meaning of the model and the interpretation. This paper adopts the two-step method to decompose the Shapley value. We predict the model residuals based on the panel regression using Eqs. (8) and (10). Then we conduct a Shapley decomposition of variable contribution based on Eqs. (9) and (11). Based on the decomposition of the $R^2$ of the regression model, we can get the contribution of different natural disasters to power outages.

$$\ln(\text{Outage frequency}_{id}) = \mathbf{X}'\theta + \delta_{iym} + \varepsilon_{idf} \quad (8)$$

$$\varepsilon_{idf} = \beta_1 \text{Strong\_wind}_{id} + \beta_2 \text{Rainstorm}_{id} + \beta_3 \text{Cold\_wave}_{id} \\ + \beta_4 \text{Geo\_hazard}_{id} + \beta_5 \text{Wildfire}_{id} + \beta_6 \text{Heatwave}_{id} + \vartheta_{id} \quad (9)$$

$$\ln\left(\text{Outage duration}_{id}\right) = \mathbf{X}'\theta + \delta_{iym} + \varepsilon_{idd} \quad (10)$$

$$\varepsilon_{idd} = \beta_1 \text{Strong\_wind}_{id} + \beta_2 \text{Rainstorm}_{id} + \beta_3 \text{Cold\_wave}_{id} \\ + \beta_4 \text{Geo\_hazard}_{id} + \beta_5 \text{Wildfire}_{id} + \beta_6 \text{Heatwave}_{id} + \vartheta_{id} \quad (11)$$

## Back-of-the-envelope calculation

The potential economic costs are calculated by the back-of-the-envelope analysis. We adopt Eqs. (12), (13) to estimate the economic

costs of those outages caused by the natural disasters.

$$\text{Total Outage hour}_{id} = \beta_1 \text{Strong\_wind}_{id} + \beta_2 \text{Rainstorm}_{id} + \beta_3 \text{Cold\_wave}_{id}$$
$$+ \beta_4 \text{Geo\_hazard}_{id} + \beta_5 \text{Wildfire}_{id} + \beta_6 \text{Heatwave}_{id} \quad (12)$$
$$+ \mathbf{X}'\theta + \delta_{iym} + \varepsilon_{id}$$

$$\text{Loss} = \sum_t \sum_i \sum_l \text{load}_l * \text{weight}_{il} * \sum_j \beta_j * \Delta \text{Natural disaster}_{jt} * \text{eleh}_i * \text{ad}$$

$$(13)$$

Where $\beta_j$ represents the $j$ disaster-induced outage coefficients from Eq. (12). $\text{load}_l$ represents the average economic loss per kilowatt-hour for types $l$ of loads (residential, industrial, commercial, and mixed-use loads). Based on the work of Zhang and Huang[36], the average economic loss for the residential load, industrial load, commercial load and mixed-use load is 13, 160, 109, and 112 RMB/kWh, respectively. $\text{weight}_{il}$ represents the proportion of outages affecting each type of load in county $i$. $\Delta \text{Naturaldisaster}_{jt}$ represents the change in the j-type of natural disaster under the Shared Socioeconomic Pathways-Representative Concentration Pathways (SSP-RCP) scenarios from the year 2022 to 2099, relative to the period 2019–2021. The $\text{eleh}_i$ is the amount of electricity consumption per hour for county $i$. ad is an average adjustment factor, and represents the number of people affected by each outage. Based on our power outage dataset, each outage affects approximately 1330 households. The average household size is 2.6 people (data sourced from the National Bureau of Statistics of China).

### Reporting summary
Further information on research design is available in the Nature Portfolio Reporting Summary linked to this article.

## Data availability
Records of daily weather conditions were retrieved from the China Meteorological Data Service Center (MDSCC). Data on wildfires and geological hazards are derived from the global natural Hazards database provided by the National Cryosphere Desert Data Center (NCDC). The raw and processed electricity outage data are web-scraped from local public utility websites. The socio-demographic data at the county level was collected from the China County Statistical Yearbook. Other data used for this study were all retrieved from publicly available sources. All data used in this study was publicly available on GitHub (https://github.com/Adaptability1/outage.git).

## Code availability
All data are processed and analyzed in Stata (16MP) and Python. The figures are produced in Stata Studio and ArcMap. The custom code is available on GitHub at https://github.com/Adaptability1/outage.git.

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

## Acknowledgements
Funding for this research was provided by the National Natural Science Foundation of China (Reference No. 72243001, 72074026, 72141302, 72222017, 72504027), the National Funded Postdoctoral Program of China (Ref.GZC20233391), China Postdoctoral Science Foundation (Reference No. 2023M740236).

## Author contributions
B.W., H.S., Y.M.Q., N.N.D., and Z.H.W. designed the study. B.W., H.S., Y.M.Q., and N.N.D. completed econometric model-related work. B.W., H.S., Y.M.Q., and N.N.D. wrote the first draft. B.W., H.S., Y.M.Q., N.N.D., D.N., X.C.S., Z.H.W.,. and Y.W. contributed to the revision of the manuscript.

## Competing interests
The authors declare no competing interests.
