## [Transparent Peer Review File · Nature Communications]

Unequal power outages induced by natural disasters

Corresponding Author: Professor Yueming Qiu

Version 0:

Reviewer comments:

Reviewer #1

(Remarks to the Author)

Through empirical analysis of the impact of disasters on poor and non-poor counties, this paper finds that poor counties are more vulnerable to power outages caused by natural disasters, and points out this difference will widen further in the future. Generally, the study is original, and has practical value to the field. However, the methods need further improvement to fully support conclusions. Comments are listed as follows.

- 1) Due to the different baseline levels of outages in non-poverty and poverty counties (in the absence of disasters), demonstrating the unequal impact of natural disasters on two areas requires not only the percentage indicators (e.g. 5.19%/3.80% increase in frequency), but also the incremental values. It is suggested to provide more comprehensive metrics for comparison.
- 2) In the comparison of disaster impacts on poor and non-poor counties, the outage duration and frequency are applied. However, the outage loss is also an important indicator in the benefit analysis for investment decisions. As important loads are likely to be heavier in non-poor areas, loss comparisons are needed.
- 3) In the Projection part, the potential economic costs are calculated by Eq. (7) based on average economic loss for residential consumers. It is recommended to account for different types of loads, including industrial, commercial, etc.
- 4) The long-term projections of climate evolution are reasonable, but the guide value of long-term projections of power outage is questionable due to the neglect of emergency response capabilities improvement and grid infrastructure upgrades. For example, some areas are significantly different in their ability to cope with wind disasters than they were 10 years ago.
- 5) There may be many stations in a county, and the outage durations varies from station to station, are there any methods to aggregate these power outage data at the county level?
- 6) The readability of the Tables should be improved. Data in Tables are not explained or annotated.

(Remarks on code availability)

Reviewer #2

(Remarks to the Author)

This manuscript describes an empirical study on how natural disasters impact power outages in China, considering the poverty distribution. The work has practical values. The reviewer has the following comments/questions:

1. The power outage data is usually restricted or sensitive data. This manuscript acquired high-frequency point-level outage data, which is from government website upon request according to the statement. However, it is not that easy especially that is related to national security. So the data availability regarding the high-resolution power outage data needs justification.
2. The paper states that the natural disasters will have more impact on poverty counties from the empirical studies. However, it does not say the frequency and intensity of natural disasters on poverty and non-poverty counties. Maybe the high impact is from more frequent occurrence of disasters, or the intensity of the disasters are higher in poverty counties. If we want to compare the impact of natural disasters on power outage frequency and intensity from the perspective of power grid features, it is fair to make the frequency and intensity of natural disasters the same.
3. There may be other issues regarding the comparison. Sometimes the natural disasters may impact a large area, where both the poverty counties and non-poverty counties are impacted. In this scenario, due to the fact that non-poverty may have higher priority, the restoration and recovery sources may be allocated to non-poverty areas, so this uneven impact may be exacerbated. How to count these coupling impacts need discussion.
4. Regarding to power outage loss, it may not be the case that the higher frequency and longer duration power outages will have higher loss. We should also consider the load priority.

(Remarks on code availability)

The reviewer thinks that the data availability regarding the high-resolution power outage data may be an issue for others to reproduce the analysis.

Version 1:

Reviewer comments:

Reviewer #1

(Remarks to the Author)

Thank you for addressing my comments point-by-point. In general, most of my concerns have been addressed to some extent, though some still require further discussion and analysis in the revised manuscript. The following comments aim to make the assumptions more practical and strengthen the conclusions:

1. For the outage loss estimation in the Methods section, the use of population-based normalization might not be appropriate for this study, as the primary focus is to analyze the relationship between natural disasters and economic development levels to power outages. Please conduct a comparative analysis, using GDP (either total or per capita) as the basis for normalization.
2. In the Results section, Beyond the general principle that "natural disasters induce more power outages," are there countries where fewer outages occur under similar natural disaster conditions? These counterintuitive observations could provide valuable insights and help decision-makers assess the robustness of the derived conclusions.
3. In the Subsection "Power outage and grid natural disaster", The relationship between natural disasters and power outages warrants further investigation, especially since power outages are not always caused by natural disasters. For example, planned outages are often published by public utility websites. It would be helpful to clarify and justify why power outages in this study are assumed to be primarily induced by natural disasters.
4. Regarding the country-level investment information, please provide the data sources and, if possible, make them open-source. This data directly impacts the conclusions in the "Power Outages Caused by Natural Disasters in Poverty-Stricken Counties" subsection. Additionally, national-level investments should also be considered, as they can influence the impacts and losses, either directly or indirectly.
5. In Subsection "Power outages caused by natural disasters in poverty counties", using the Shapley value to decompose the impact of disasters on power outages should be further validated, particularly since rainstorms and strong winds are often highly correlated during typhoon events. The Shapley value assumes that contributions can be fairly separated, but these disasters typically occur in a dependent or sequential pattern, which may complicate the application of this method.

(Remarks on code availability)

Natural disaster and power outage information are published, while the country-level investment information is missing. The correlation among natural disasters should be analyzed when the Sharly value is adopted.

Reviewer #2

(Remarks to the Author)

The revised version has addressed my comments.

(Remarks on code availability)

Version 2:

Reviewer comments:

Reviewer #1

(Remarks to the Author)

Thank you for your point-by-point responses to my previous comments. Some issues still require further discussion and analysis in the revised manuscript:

1. The introduction lacks proper citations for certain data points, such as the percentage of power outages caused by natural factors in China. While you mentioned the "2021 China Power Reliability Annual Report" in your response, to my knowledge, this report doesn't clearly define what constitutes "natural disasters." In practice, outages caused by extreme weather events are typically categorized and counted separately as major incidents rather than being included in routine reliability statistics for distribution networks. The "natural factors" in reliability statistics usually refer to more common occurrences like tree branch contacts. Could you please further discuss the correlation between the outage dataset and natural disaster dataset used in your study?
2. The outage examples provided in the introduction (e.g., cold waves and Sichuan-Chongqing power restrictions) don't strongly relate to your core theme. These were large-scale events that significantly affected both urban and non-poor areas. I recommend focusing more on examples that highlight the unique characteristics and challenges of power outages in impoverished regions. Additionally, your result figures show a notable spatial discrepancy between disaster distributions and poverty-stricken areas, which seems counterintuitive to your main conclusions.

3. The statement (lines 66-69) suggesting limited research on natural disasters' impact on power infrastructure resilience is inaccurate. This field has actually been studied for decades with substantial scholarly work. Please revise this section to provide a more comprehensive and precise discussion of existing research and its limitations.
4. Figures 1 and 3 contain extensive white areas - do these represent missing outage data? If so, this could significantly impact your results as many impoverished counties fall within these areas. Moreover, the presented data doesn't visually support your conclusion that "power outages are more severe in poor counties." Please address this discrepancy.
5. The presentation format and information included in Table 1 are currently confusing and need revision for better clarity.
6. Please provide justification for using datasets from different years (2013 poverty data vs. 2020 outage data) and discuss how this temporal discrepancy might affect your analysis.

(Remarks on code availability)

Reviewers comments:

Reviewer #1 (Remarks to the Author):

Comments: Through empirical analysis of the impact of disasters on poor and non-poor counties, this paper finds that poor counties are more vulnerable to power outages caused by natural disasters, and points out this difference will widen further in the future. Generally, the study is original, and has practical value to the field. However, the methods need further improvement to fully support conclusions. Comments are listed as follows.

Response:

We sincerely thank the reviewer for the remarks on the overall work. We are grateful for the constructive feedback and the opportunity to further improve our work. We have made every effort in improving the quality of our research based on your suggestions. The revised manuscript is highlighted in blue. Our point-by-point responses are provided below.

Comments:(1) Due to the different baseline levels of outages in non-poverty and poverty counties (in the absence of disasters), demonstrating the unequal impact of natural disasters on two areas requires not only the percentage indicators (e.g. 5.19%/3.80% increase in frequency), but also the incremental values. It is suggested to provide more comprehensive metrics for comparison.

Response:

Thanks for the suggestion about the necessity of providing the impact of natural disasters on incremental values of power outages to demonstrate the unequal impact of natural disasters on poverty and non-poverty counties. Following the advice of the reviewers, we have incorporated different measurement approaches for the variables here to provide more comprehensive metrics for comparison. We have primarily supplemented our analysis with the following two methods:

First, we estimate the impact of natural disasters on incremental values of power outages for poverty and non-poverty counties based on Equation (1)-(4) using the

absolute value (rather than taking the logarithm form) of the outage frequency and outage duration. In these Equations, we are interested in the parameter α_1 , which measures the incremental values of the frequency or duration of power outages associated with a one-unit change in the natural disasters.

$$Outage_frequency_{id} = \alpha_1 Natural\ disaster_{id} + X'\theta + \delta_{iym} + \varepsilon_{id} \quad (1)$$

$$Outage_duration_{id} = \alpha_1 Natural\ disaster_{id} + X'\theta + \delta_{iym} + \varepsilon_{id} \quad (2)$$

$$Outage_frequency_{id} = \alpha_1 Natural\ disaster_{id} + \alpha_2 Natural\ disaster_{id} * Poverty_i + X'\theta + \delta_{iym} + \varepsilon_{id} \quad (3)$$

$$Outage_duration_{id} = \alpha_1 Natural\ disaster_{id} + \alpha_2 Natural\ disaster_{id} * Poverty_i + X'\theta + \delta_{iym} + \varepsilon_{id} \quad (4)$$

Where $Outage_frequency_{id}$ denotes the power outage frequency on day d in county i.

$Outage_duration_{id}$ denotes the power outage duration on day d in county i.

$Natural\ disaster_{id}$ represents the cumulative occurrence of natural disasters including wildfires, geological disasters, heatwaves, cold waves, strong winds and rainstorms on day d in county i. The coefficient α_1 represents incremental values of power outage frequency or duration with one unit change in natural disasters. X control for the holiday and weekend dummy variables. The county-month-by-year fixed effects (δ_{iym}) are included as needed to account for time changes in each county such as cultural economic conditions, energy transition policies, and local infrastructure.

The regression results are shown in Table R1. We can see that grid-related natural disasters have unequal influences on power outages across poverty versus non-poverty counties, indicating the robustness of our main results. Specifically, the results in Columns (2) and (3) show that each additional unit of natural disasters increases the frequency of power outages by 0.0051 in poverty counties and by 0.0028 in non-poverty counties. The results for the interaction term in Column (4) indicate that the difference is statistically significant. Similarly, the results in Columns (6) and (7) show that each

additional unit of natural disasters increases the duration of each power outage by 0.6929 hours in poverty counties and by 0.2569 hours in non-poverty counties. The results for the interaction term in Column (8) indicate that this difference is statistically significant.

Table R1 Impacts of grid-related natural disasters on incremental values of power outages

	Outage frequency				Outage duration			
	(1)	(2)	(3)	(4)	(5)	(6)	(7)	(8)
	ALL	Poverty	Non_Poverty	ALL	ALL	Poverty	Non_Poverty	ALL
Natural disaster	0.0031*** (0.0003)	0.0051*** (0.0008)	0.0028*** (0.0003)	0.0028*** (0.0003)	0.3257*** (0.0452)	0.6929*** (0.1759)	0.2569*** (0.0427)	0.2565*** (0.0427)
Natural disaster#				0.0023*** (0.0008)				0.4449** (0.1808)
Control	YES	YES	YES	YES	YES	YES	YES	YES
County*YM	YES	YES	YES	YES	YES	YES	YES	YES
Observation	1497450	306190	1191260	1497450	1504150	306190	1197960	1504150
R2	0.1765	0.1381	0.1914	0.1765	0.1410	0.1144	0.1550	0.1410

Notes: *p<0.10, **p<0.05, *** p<0.01. Standard errors are clustered at the county level and reported below the coefficients. Control variables include dummy variables for holidays and weekends. County*YM is the county-month-by-year fixed effect.

Second, in addition to the absolute values of outage frequency and outage duration, we have constructed a more comprehensive indicator (*Total_outage_hours*), which captures both the frequency and duration characteristics of power outages. The results are presented in Table R2. As shown, despite employing different indicators for power outages, it is evident that the impact of natural disasters on poverty counties is significantly greater than on non-poverty counties, aligning with our primary findings

and demonstrating the robustness of our results. To avoid redundancy, detailed results for Table R2 can be found in our response to Comment (2).

Overall, as suggested by the reviewer, we have provided more comprehensive metrics for comparison by introducing absolute value (rather than taking the logarithm form) of the outage frequency, outage duration and *Total_outage_hours*. Once again, we appreciate the valuable suggestion and hope our response has adequately addressed the reviewer's concerns. In the revised manuscript, we have added the details in the Results Section (Page 7, lines 18-25).

Considering the different baseline levels of outages in different counties, we examine the impact of natural disasters on incremental values of power outages (Supplementary Table S2 and Table S3). The results show that poverty counties experience more outage frequency and longer outage duration caused by natural disasters, showing the robustness of the results. We also construct a more comprehensive indicator (Total_outage_hours), which simultaneously captures the frequency and duration characteristics of power outages (see Methods section). The results are highly consistent with baseline results (Supplementary Table S4).

Comments:(2) In the comparison of disaster impacts on poor and non-poor counties, the outage duration and frequency are applied. However, the outage loss is also an important indicator in the benefit analysis for investment decisions. As important loads are likely to be heavier in non-poor areas, loss comparisons are needed.

Response:

We appreciate the reviewer for this constructive suggestion regarding outage loss. Given that we do not have detailed loss information for each power outage, we calculate the marginal economic losses caused by power outages due to natural disasters for both poverty and non-poverty counties to compare the impact of natural disasters on outage loss. We have addressed this concern through the following three steps. Firstly, to measure the marginal losses of disaster impacts on poverty and non-poverty counties, we construct an indicator (*Total_outage_hours*) to simultaneously capture both outage

frequency and duration characteristics. Subsequently, we estimate the marginal impact of natural disasters on *Total_outage_hours* for poverty and non-poverty counties. Finally, based on the average proportion of outages impacting various loads and corresponding economic losses for each load in both poverty and non-poverty counties, the marginal loss in both poverty and non-poverty counties has been calculated. The details are in the following:

First, to capture a comprehensive impact of natural disasters on power outages, we construct an indicator (*Total_outage_hours*), which is designed to simultaneously capture outage frequency and duration characteristics. *Total_outage_hours* equals per capita outage frequency (*Outage_frequency*) in the county on that day multiplied by the average duration per outage (*Outage_duration*). As an example, we use the power outage situation in Pianguan County, Shanxi, on October 15th, 2021, to illustrate the calculation steps for *Total_outage_hours*. Pianguan County experienced two power outages on that day: the first lasting 2.75 hours and the second lasting 7.5 hours. The total population of Pianguan county is 7.3382 (in ten thousand units). The *Total_outage_hours* equals to 1.40 ($2/7.3382*(2.75+7.5)/2$).

Second, we employ Equation (5) to estimate the marginal impact of natural disasters on *Total_outage_hours* in poverty and non-poverty counties.

$$Total_outage_hour_{id} = \alpha_1 Natural_disaster_{id} + X'\theta + \delta_{iym} + \varepsilon_{id} \quad (5)$$

The parameters and key variables are interpreted in the same way as described above. The regression results are shown in Table R2. The estimated parameters (α_1) for poverty counties and non-poverty counties are 0.1230 and 0.0358, respectively, indicating that the impact of natural disasters on poverty counties is greater than that on non-poverty counties. The results for the interaction term in Column (4) indicate that the difference is statistically significant.

Table R2 Impacts of grid-related natural disasters on total power outage hours

	Total_outage_hours			
	(1)	(2)	(3)	(4)
	ALL	Poverty	Non_Poverty	ALL
Natural disaster	0.0494*** (0.0074)	0.1230*** (0.0338)	0.0358*** (0.0062)	0.0356*** (0.0062)
Natural disaster*Poverty				0.0882** (0.0344)
Control	YES	YES	YES	YES
County*YM	YES	YES	YES	YES
Observation	1497450	306190	1191260	1497450
R2	0.0868	0.0967	0.0804	0.0868

Notes: *p<0.10, **p<0.05, *** p<0.01. Standard errors are clustered at the county level and reported below the coefficients. Control variables include dummy variables for holidays and weekends. County*YM is the county-month-by-year fixed effect.

Finally, we estimate the marginal economic losses caused by power outages for poverty and non-poverty counties using Equation (6), by considering the average proportion of outages affecting various loads for poverty and non-poverty counties and the corresponding losses for each load.

$$Loss_{-m_p} = \alpha_{1p} * \sum_l load_l * weight_{lp} * ele_{-h_p} * ad \quad (6)$$

α_{1p} is the estimated parameter obtained from Equation (5), indicating the marginal impact of natural disasters on daily total outage hours per person for poverty and non-poverty counties. p measures whether the county is poor or not. $load_l$ represents the average economic loss per kilowatt-hour for types l of loads (residential, industrial commercial, and mixed-used loads). Based on the work of Zhang and Huang (2022), the average economic loss for the residential load, industrial load, commercial load, and mixed-used load is 13, 160, 109, and 112 RMB/kWh, respectively. $weight_{lp}$

represents the proportion of outages affecting types l of loads for poverty and non-poverty counties. The ele_h_p is the average electricity consumption per hour for poverty and non-poverty counties. ad is an average adjustment factor, and represents the number of people affected by each outage. Based on our power outage dataset, each outage affects approximately 1,330 households. The average household size is 2.6 people (data sourced from the National Bureau of Statistics of China). Therefore, after considering different types of loads, the marginal economic losses caused by natural disasters in poverty counties are approximately 1.56 times those in non-poverty counties.

Overall, we are deeply grateful for the reviewer's suggestion to further distinguish between the loads of poverty counties and non-poverty counties for outage loss comparisons, which has enriched our research findings. Accordingly, we have incorporated these detailed comparisons in our revised manuscript, details can be found in the Results Section (Page 7, lines 26-32).

Considering the importance of outage loss, we further compare the marginal economic losses caused by power outages due to natural disasters in poverty and non-poverty counties by the average proportion of outages impacting various loads in poverty and non-poverty counties and the corresponding economic losses for each load. We find that the marginal losses in poverty counties are 1.56 times higher than those in non-poverty counties, based on the estimated parameters of natural disasters on Total_outage_hours (Supplementary Table S4).

Reference

Zhang, W., and Huang, Z. (2022). Research on priority strategy of overhaul without power interruption based on user outage loss assessment (in Chinese). *Electric Engineering 01*, DOI:10.19768/j.cnki.dgjs.

Comments:(3) In the Projection part, the potential economic costs are calculated by Eq. (7) based on average economic loss for residential consumers. It is recommended to

account for different types of loads, including industrial, commercial, etc.

Response:

We appreciate the reviewer providing this suggestion. Accounting for different types of loads could further enhance the rigor of our research findings. Following the reviewer’s suggestion, we have recalculated the potential economic losses from power outages caused by natural disasters, taking into account different types of loads. We have addressed this issue through the following three aspects.

First, to capture a comprehensive impact of different natural disasters on power outages, we construct an indicator (*Total_outage_hours*), which is designed to simultaneously capture outage frequency and duration characteristics. *Total_outage_hours* equals per capita outage frequency (*Outage_frequency*) in the county on that day multiplied by the average duration per outage (*Outage_duration*).

Second, We estimate the impacts of different natural disasters on *Total outage hours* based on Equation (7), the results are shown in Table R3. Specifically, in poverty counties, the strong wind, rainstorms, cold waves and heatwave increase total outage hour by 0.0693, 0.9272, 0.0578 and 0.0206, respectively, compared to 0.0425, 0.1710, 0.0120 and 0.0127 in non-poverty counties.

$$Total_Outage_hour_{id} = \beta_1 Strong\ wind_{id} + \beta_2 Rainstorm_{id} + \beta_3 Cold\ wave_{id} + \beta_4 Geo\ hazard_{id} + \beta_5 Wildfire_{id} + \beta_6 Heatwave_{id} + X'\theta + \delta_{iym} + \varepsilon_{id} \quad (7)$$

In Equation (11), β_j represents the impact of j natural disaster on power outage. The variables are interpreted in the same manner as previously described.

Then we account for different types of loads (including residential, industrial commercial, and mixed-used loads) to project the economic losses caused by power outages in Equation (8). The details of the Equations are as follows:

$$Loss = \sum_i \sum_i \sum_i load_l * weight_{il} * \sum_j \beta_j * \Delta Nature\ disaster_{jt} * ele_h_i * ad \quad (8)$$

In Equation (8), $load_l$ represents the average economic loss per kilowatt-hour for types

l of loads (residential, industrial commercial, and mixed-used loads). Based on the work of Zhang and Huang (2022), the average economic loss for the residential load, industrial load, commercial load, and mixed-used load is 13, 160, 109, and 112 RMB/kWh, respectively. $weight_{il}$ represents the proportion of each type of load in county i affected by outages. $\Delta Nature\ disaster_{jt}$ represents the change in the j -th type of natural disaster under the Shared Socioeconomic Pathways-Representative Concentration Pathways (SSP-RCP) scenarios from the year 2022 to 2100, relative to the period 2019-2021. The ele_h_i is the electricity consumption per hour for county i . ad is an average adjustment factor, and represents the number of people affected by each outage.

Based on our calculations, the projected damages between 2022 and 2100 are estimated to range from 549 to 1340 billion RMB. The increase in future disasters is projected to widen the outage gaps caused by natural disasters between poverty and non-poverty counties under the SSP-RCP126 (157.82%) and SSP-RCP585 (157.76%) scenarios.

Table R3 Impacts of different grid-related natural disasters on total outage hours

	Total_outage_hours	
	(1)	(2)
	Poverty	Non_Poverty
Strong_wind	0.0693*** (0.0258)	0.0425*** (0.0144)
Rainstorm	0.9272*** (0.2966)	0.1710*** (0.0463)
Cold_wave	0.0587** (0.0271)	0.0120*** (0.0044)
Geo_hazard	0.0578 (0.0410)	0.0130 (0.0099)

Wildfire	0.1653 (0.1278)	-0.0153 (0.0103)
Heatwave	0.0206 (0.0159)	0.0127*** (0.0030)
Control	YES	YES
County*YM	YES	YES
Observation	306190	1191260
R2	0.0975	0.0805

Notes: *p<0.10, **p<0.05, *** p<0.01. Standard errors are clustered at the county level and reported below the coefficients. Controls include holiday and weekend dummy variables. County*YM is the county-month-by-year fixed effect. Observation is the sample size. R2 represents the goodness-of-fit of the regressions.

Once again, we express our gratitude for the reviewer's valuable suggestions. Inspired by the reviewer, we have taken into account various types of loads, including residential load, industrial load, commercial load, and mixed-used load in our analysis. The details of these adjustments have been incorporated into the Results Section of the revised manuscript (Page 10, lines 42-44; Page 11, lines 1-7).

The projected increase in natural disasters is expected to result in 0.21 h/person/year from 2022 to 2100 under the SSP-RCP126 scenario, and 0.41 h/person/year under the SSP-RCP585 scenario. We also perform a back-of-the-envelope calculation of the economic cost of power outage hours induced by natural disasters (see Methods). Based on our calculations, the projected damages from 2022 to 2100 are estimated to range from 549 to 1340 billion RMB. We find that climate change will further widen the outage gaps caused by natural disasters between poverty and non-poverty counties in SSP-RCP126 (157.82%) and SSP-RCP585 scenarios (157.76%). These findings indicate that the reliability of power supply in poverty counties will face substantial challenges in the future.

Reference

Zhang, W., and Huang, Z. (2022). Research on priority strategy of overhaul without power interruption based on user outage loss assessment (in Chinese). *Electric Engineering 01*, DOI:10.19768/j.cnki.dgjs.

Comments: (4) The long-term projections of climate evolution are reasonable, but the guide value of long-term projections of power outage is questionable due to the neglect of emergency response capabilities improvement and grid infrastructure upgrades. For example, some areas are significantly different in their ability to cope with wind disasters than they were 10 years ago.

Response:

Thank you for this insightful comment. We acknowledge the limitation in not fully considering the corresponding impact of emergency response capabilities improvement and grid infrastructure upgrades on power outages caused by natural disasters. Although we cannot ideally depict future improvements, we have still explored this issue through assumptions in several scenarios. This will help us better understand the role of future grid infrastructure upgrades in mitigating this issue. Specifically, we have constructed the following three scenarios:

(1) the Reference Scenario, where the grid infrastructure remains unchanged; (2) the Improvement to Non-Poverty Counties Level Scenario, where the impact of natural disasters on power outages in all poverty counties is projected to decrease to the current level experienced by non-poverty counties in China by 2060; (3) the Improvement to the Top 20% Level Scenario, where the impact of natural disasters on power outages in all counties is projected to reduce to the current level experienced by the top 20% of counties with the highest power investment in China by 2060. In the last two scenarios, considering China's goal to establish a new energy system by 2060—where China will vigorously improve the comprehensive regulation capability of the power system and accelerate the construction of flexible power regulation to build a strong smart grid and enhance grid security—we assume that counties currently below the target level will gradually reduce the impact of natural disasters on power outages at a steady rate until 2060. After 2060, this impact is expected to stabilize at the target level.

Based on the three scenarios established for capabilities improvement and grid infrastructure upgrades, we re-estimate the impact of future changes in natural disasters on power outages under different scenarios. The results are shown in the Table R4. The results indicate that the improvement of emergency response capabilities and grid infrastructure significantly reduces the increase of outage hours caused by the changes in natural disasters compared to the Reference Scenarios. Specifically, in SSP-RCP126, the change of natural disasters will increase the outage hours by 0.21 h/person/year in Reference Scenarios, while 0.03 h/person/year in Improvement to the Non-poverty Counties Level Scenario and 0.01 h/person/year in Improvement to the Top 20% Level Scenario. In SSP-RCP585, the change of natural disasters will increase the outage hours by 0.41 h/person/year in Reference Scenarios, compared to 0.22 h/person/year in Improvement to the Non-poverty Counties Level Scenario and 0.18 h/person/year in Improvement to the Top 20% Level Scenario. Furthermore, the gaps between poverty and non-poverty counties in the Improvement to the Non-Poverty Level Scenario and the Improvement to the Top 20% Level Scenario reduce under SSP-RCP126 (-54.33% and -53.90%) and SSP-RCP585 (-67.78% and -66.65%) compared to the Reference Scenarios. These findings suggest that the enhancements in emergency response capabilities and grid infrastructure are significantly important in mitigating natural disaster-induced outages and reducing the gaps between poverty and non-poverty counties.

Table R4 Impacts of future natural disasters on outage hours in three scenarios

	SSP-RCP126		SSP-RCP585	
	Δ Outage hours	Δ Gaps	Δ Outage hours	Δ Gaps
	(h/person/year)		(h/person/year)	
Reference	0.21	157.82%	0.41	157.76%

Improvement to the				
Non-poverty	0.03	-54.33%	0.22	-67.78%
Counties Level				
Improvement to				
the Top 20% Level	0.01	-53.90%	0.18	-66.65%

Once again, we are grateful for the reviewer's suggestions. By constructing new scenarios, we have further strengthened and quantified the role of enhancements in emergency response capabilities and grid infrastructure upgrades in mitigating the issue. In the revised manuscript, we have added the description in the Results Section (Page 11, lines 9-21). For a comprehensive review of the detailed revisions, please check the "Projections under different scenarios" section in the Supplementary Material.

Additionally, in the Supplementary section titled "Projections under different scenarios", we explore the potential impact of improvements in emergency response capabilities and grid infrastructure upgrades on power outages caused by natural disasters. Apart from the Reference Scenario, we also construct two scenarios: the Improvement to the Non-Poverty Counties Level Scenario and the Improvement to the Top 20% Level Scenario. The results indicate that the improvement of emergency response capabilities and grid infrastructure significantly reduces the increase of outage hours caused by the changes in natural disasters compared to the Reference Scenarios. Furthermore, the gaps between poverty and non-poverty counties in the Improvement to Non-Poverty Counties Level Scenario and the Improvement to the Top 20% Level Scenario also reduce compared to the Reference Scenarios (see Table S11). This estimation provides us with an indication of the potential damage that can be mitigated through investments in enhanced power systems.

Comments: (5) There may be many stations in a county, and the outage durations varies from station to station, are there any methods to aggregate these power outage data at the county level?

Response:

We thank the reviewer for raising this concern. In the revised manuscript, we have provided a more transparent and detailed explanation regarding the aggregation method in the Methods section (Pages 14, lines 35-44; Pages 15, lines 1-8). The specific description in the manuscript is as follows:

There are many outage records across different locations within a county for a given date. Since the population varies significantly across counties, the number of outage records may correlate with population size. To address this issue, we calculate the daily outage frequency by aggregating the total number of outages within a county and normalizing it by the county's population. The outage duration for a given day is calculated by averaging the hours of all outage records within the county. In addition, we constructed an indicator (Total_outage_hours) to simultaneously capture both outage frequency and duration characteristics. This indicator equals the per capita outage frequency (Outage_frequency) in the county on that day multiplied by the average duration per outage (Outage_duration). As an example, we use the power outage situation in Pianguan County, Shanxi, on October 15th, 2021, to illustrate the calculation steps for Outage frequency, Outage duration and Total_outage_hours. Pianguan County experienced two power outages on that day: the first one lasting 2.75 hours, and the second one lasting 7.5 hours. The total population of Pianguan County is 7.3382(in ten thousand units). The Outage_frequency is calculated to be 0.27 (2/7.3382). And the Outage_duration is calculated to be 5.125 ((2.75+7.5)/2) hours. The Total_outage_hours equals to 1.40 (2/7.3382(2.75+7.5)/2).*

We hope that our explanation above addresses any confusion regarding the aggregation of data at the county level for both the reviewer and the readers. We are once again grateful for the reviewer's suggestions, which have significantly enhanced the clarity and quality of our manuscript.

Comments:(6) The readability of the Tables should be improved. Data in Tables are not explained or annotated.

Response:

We appreciate the reviewer's feedback regarding the readability of the tables in our previous manuscript. In the revised manuscript, we have enhanced the readability of each table by ensuring consistent formatting, refining alignment, and adding clear labels for all variables and units where applicable. Additionally, we have included detailed annotations to clarify the data and explain each variable's meaning in the table notes, and provided more detailed interpretations of the regression coefficients in the Results section.

We have provided a revised interpretation of Table 1 below (Page 5, lines 28-43; Page 6, lines 1-23; and Page 7, lines 1-4).

*We explore the impact of natural disasters on power outages using a two-way fixed effects model. The regression results in Table 1 indicate that, in all model specifications, natural disasters have a statistically significant and positive impact on power outages. Specifically, natural disasters increase outage frequency by 4.04% (Column 1, Table 1) and extend outage duration by 5.93% (Column 2, Table 1) on average for an additional unit increase in natural disasters. All models' specifications include county-month-by-year fixed effect (County*YM), which captures the time-varying characteristics in each county such as cultural economic conditions, energy transition policies and local infrastructure. We cluster the standard errors at the county level. We further explore the impacts of different types of disasters on power outages. We find that strong wind, rainstorms, cold waves, geological hazards, and heatwaves significantly increase outage frequency by 2.98%, 7.99%, 3.77%, 4.45%, and 3.60% (Column 3, Table 1), respectively. Additionally, these natural disasters increase in outage duration by 4.38%, 11.63%, 6.63%, 5.98%, and 4.99% (Column 4, Table 1), respectively.*

Among these natural disasters, rainstorms have the most significant influence on both the outage frequency and duration (Columns 3 and 4, Table 1). It must be emphasized that the observed impact of rainstorms on power outages could be a result of the combined effects of rainstorms and lightning due to the collinearity issues between these two factors. Rainstorms and lightning inflict on electrical equipment and disrupt the power supply network, thereby exacerbating the vulnerability

to power outages. Notably, we find no substantial impact from wildfires. This may be due to the relatively infrequent occurrence and moderate severity of wildfires in China. Based on the wildfire data in 2020, the United States experienced 51 times more forest fires than China, with an economic loss 161 times higher (MEMPRC, 2021; NIFC, 2023). To further eliminate the influence stemming from other confounding factors, we conducted a reanalysis controlling for, and thus excluding, the central environmental protection inspections (CEPI), coal price fluctuations, and the impact of COVID-19 on power outages (Supplementary Table S5). The reanalysis results are consistent with our main results, indicating that all grid-related natural disasters, except wildfires, result in increased outages.

Table 1. Impacts of grid-related natural disasters on power outages

	(1)	(2)	(3)	(4)
	ln(Outage frequency)	ln(Outage duration)	ln(Outage frequency)	ln(Outage duration)
Natural disaster	0.0404*** (0.0029)	0.0593*** (0.0055)		
Strong wind			0.0298*** (0.0058)	0.0438*** (0.0108)
Rainstorm			0.0799*** (0.0088)	0.1163*** (0.0162)
Cold wave			0.0377*** (0.0046)	0.0663*** (0.0095)
Geo hazard			0.0445*** (0.0103)	0.0598*** (0.0196)
Wildfire			0.0059 (0.0131)	0.0414 (0.0330)
Heatwave			0.0360*** (0.0042)	0.0499*** (0.0082)
Control	YES	YES	YES	YES

County*YM	YES	YES	YES	YES
Observation	1497450	1504150	1497450	1504150
R2	0.8410	0.2505	0.8410	0.2505

*Notes: The statistical significance of the coefficients is indicated by asterisks: * denotes $p < 0.10$, ** denotes $p < 0.05$, and *** denotes $p < 0.01$. Standard errors are clustered at the county level and reported below the coefficients. $\ln(\text{Outage frequency})$ refers to the natural logarithm of power outage frequency. $\ln(\text{Outage duration})$ refers to the natural logarithm of power outage duration. *Natural disaster* represents the intensity of natural disasters, which is the sum of *Strong wind*, *Rainstrom*, *Cold wave*, *Geo hazard*, *Wildfire* and *Heatwave*. *Strong wind* (*Rainstrom*, *Cold wave*, *Geo hazard*, *Wildfire* and *Heatwave*) is a dummy variable that equals 1 when the county experiences strong winds (rainstorms, cold waves, geological disasters, wildfire and heatwaves). Controls include dummy variables for holidays and weekends. *County#YM* is the county-month-by-year fixed effect. *Observation* refers to the sample size. *R2* indicates the goodness-of-fit of the regressions.*

Reviewer #2 (Remarks to the Author):

Comments: This manuscript describes an empirical study on how natural disasters impact power outages in China, considering the poverty distribution. The work has practical values. The reviewer has the following comments/questions:

Response:

We are grateful for the encouraging feedback and the opportunity to further enhance the quality of our work. We have made every effort to address the concerns raised by reviewer. The revised manuscript is highlighted in blue. Our point-by-point responses are provided below.

Comments:1. The power outage data is usually restricted or sensitive data. This manuscript acquired high-frequency point-level outage data, which is from a government website upon request according to the statement. However, it is not that easy especially that is related to national security. So the data availability regarding the high-resolution power outage data needs justification.

Response:

We thank the reviewer for raising the concern regarding the availability of power outage data. We acknowledge the sensitivity inherent in electricity data, particularly high-precision consumption data, which typically remains undisclosed due to its implications for commercial confidentiality and competitive advantage. However, power outage data, given its significant impact on both residential and industrial/commercial activities, is made publicly available (though they are not being systematically collected and stored in a central database). To address this concern, we have further justified the availability regarding the high-resolution power outage data from the following two aspects.

From a regulatory requirement perspective, according to the "Electricity Supply Supervision Measures" (State Electricity Regulatory Commission Order No. 27), users need to be notified of power outages due to the planning maintenance and temporary maintenance of power supply facilities. Similarly, 'Electric Power Enterprise Information Disclosure Regulations' mandate that power enterprises disclose relevant

information, including power outage notifications. These regulations are designed to protect consumers' right to know, ensure the transparency of electricity supply, and provide users with ample time to respond to power outages.

From the practical perspective of data accessibility, the local public utility websites will provide real-time power outage information and estimated recovery time, including planned outages and unplanned outages. We could obtain high-resolution power outage information, such as the start and end time, addresses and areas affected, and causes of power outages, on public utility websites, such as <https://95598.impc.com.cn/infomation/tdtz>. Third-party websites of <http://www.sttcq.com/td/> compile power outage information across China. The information in Figure R1 displays a screenshot of a power outage record from Huoqiu County, Liuan City. By tracking and scraping real-time outage information from these websites, we were able to construct the outage indicators used in this study.

[Figure Redacted]

(a)

(b)

Figure R1. A screenshot of power outage record for Huoqiu County, Liuan City, China

Note: Panel (a) illustrates the original power outage record in Chinese for Huoqiu county, Liuan City, China.

Panel (b) presents the translation power outage record in English for the same location.

Once again, we appreciate the valuable suggestion and hope our response has adequately addressed the reviewer's concerns. In the revised manuscript, we have added the description of data availability in the Method Section (Page 11, lines 14-24).

The high-resolution power outage data in China were collected from local public utility websites. Given the significant impact of power outages on both residential and industrial/commercial activities, the power outage data is made publicly available (though they are not being systematically collected and stored in a central database), and the users can get the real-time power outage information from the local public utility websites. We aggregated power outage data at the county and daily levels, covering the period from November 2019 to September 2021. The outage data includes the location of the power outage, starting time, ending time and type of power outage (planned or unplanned). Here, planned outages are notified in advance. Unplanned power outages are unexpected power failures, that may be caused by natural and man-made factors (Babar et al., 2022; Waseem and Manshadi, 2020).

Comments: 2. The paper states that the natural disasters will have more impact on poverty counties from the empirical studies. However, it does not say the frequency and intensity of natural disasters on poverty and non-poverty counties. Maybe the high impact is from more frequent occurrence of disasters, or the intensity of the disasters are higher in poverty counties. If we want to compare the impact of natural disasters on power outage frequency and intensity from the perspective of power grid features, it is fair to make the frequency and intensity of natural disasters the same.

Response:

We appreciate the reviewer's suggestion to examine whether the observed higher impact of natural disasters on power outages in poverty counties results from differences in disaster frequency or intensity. We would like to address this issue from the following two aspects.

Firstly, inspired by the reviewer's comments, we present the average number of natural disasters per day in both poverty and non-poverty counties, and then compare their differences using a T-test. The results, as shown in Table R6, indicate that poverty counties did not experience a more frequent occurrence of disasters.

Table R6 Between-group difference test of natural disasters

	Non_Poverty Counties	Poverty Counties	T test
Natural disaster	0.0274	0.0193	.0080977***
Strong wind	0.0033	0.0026	.0006659***
Rainstorm	0.0029	0.0016	.0013365***
Cold wave	0.0029	0.0020	.0009058***
Geo hazard	0.0010	0.0015	-.0004482***
Wildfire	0.0004	0.0005	-.0003942***
Heatwave	0.0169	0.0111	.0058152***

Notes: * $p < 0.10$, ** $p < 0.05$, *** $p < 0.01$.

Secondly, to compare the impact of natural disasters on power outages across varying intensities, we categorize the cumulative number of natural disasters during the study period into three groups: 0-10, 10-20, and more than 20 occurrences. We then reran the regression models within each group (as the reviewer suggested 'to make the frequency and intensity of natural disasters relatively comparable'), with results presented in Table R7. We are interested in the parameter of Natural_disaster #Poverty, which measures whether there are significant differences in the impact of natural disasters on power outages between poverty and non-poverty counties.

We find that the impact of natural disasters on power outages is significantly greater in poverty counties compared to non-poverty counties when the regions experienced 0-10 and 10-20 natural disasters in our study period. And there is no significant difference in the impact between poverty and non-poverty counties when the regions experience more than 20 natural disasters. Overall, after controlling for variations in natural disaster frequency and intensity across poverty and non-poverty counties, natural disasters still have a greater impact on poverty counties, although those regions experiencing more than 20 natural disasters show no significant differences.

Table R7: Impacts of grid-related natural disasters on power outages across different

intensity levels of natural disasters

	ln(Outage frequency)			ln(Outage duration)		
	(1)	(2)	(3)	(4)	(5)	(6)
	0-10	10-20	>20	0-10	10-20	>20
Natural_disaster	0.0469***	0.0299**	-0.0112	0.0742***	0.0684**	-0.0005
#Poverty	(0.0129)	(0.0149)	(0.0135)	(0.0251)	(0.0309)	(0.0287)
Natural_disaster	0.0055	0.0521***	0.0384***	0.0140	0.0673***	0.0548***
	(0.0048)	(0.0060)	(0.0043)	(0.0102)	(0.0113)	(0.0082)
Control	YES	YES	YES	YES	YES	YES
County*YM	YES	YES	YES	YES	YES	YES
Observation	644540	471010	381900	647890	473690	382570
R2	0.8992	0.7948	0.7797	0.2059	0.2626	0.2609

Notes: * $p < 0.10$, ** $p < 0.05$, *** $p < 0.01$. Standard errors are clustered at the county level and reported below the coefficients. Controls include dummy variables for *holidays and weekends*. County*YM is the county-month-by-year fixed effect.

Comments: 3. There may be other issues regarding the comparison. Sometimes the natural disasters may impact a large area, where both the poverty counties and non-poverty counties are impacted. In this scenario, due to the fact that non-poverty may have higher priority, the restoration and recovery sources may be allocated to non-poverty areas, so this uneven impact may be exacerbated. How to count these coupling impacts need discussion.

Response:

We thank the reviewer for the insightful comment. This is indeed a thought-provoking question that warrants further exploration. Given the limitations in power grid resources, there may be a possibility of resource allocation bias between poverty and non-poverty counties within the same city. Typically, a city, which includes multiple counties, has a higher-level power grid company that manages the grids at the county level.

To address this issue, as inspired by the reviewer, we incorporate a dummy variable ($Both_i$) in the regression model to indicate whether poverty and non-poverty counties within the same city experience natural disasters simultaneously, as shown in Equation (9)-(10).

$$\ln(Outage_frequency_{id}) = \alpha_1 Natural_disaster_{id} * Poverty_i * both_{id} + \alpha_2 Natural_disaster_{id} * Poverty_i + \alpha_3 Natural_disaster_{id} * both_{id} + \alpha_4 both_{id} * Poverty_i + \alpha_5 Natural_disaster_{id} + \alpha_6 both_{id} + X'\theta + \delta_{iym} + \varepsilon_{id} \quad (9)$$

$$\ln(Outage_duration_{id}) = \alpha_1 Natural_disaster_{id} * Poverty_i * both_{id} + \alpha_2 Natural_disaster_{id} * Poverty_i + \alpha_3 Natural_disaster_{id} * both_{id} + \alpha_4 both_{id} * Poverty_i + \alpha_5 Natural_disaster_{id} + \alpha_6 both_{id} + X'\theta + \delta_{iym} + \varepsilon_{id} \quad (10)$$

We are interested in the parameter α_1 , which measures whether the difference in the impact of natural disasters on power outages in poverty and non-poverty counties is greater when the natural disasters occur simultaneously in poverty and non-poverty counties. X control the holiday and weekend dummy variables. The county-month-by-year fixed effects (δ_{iym}) are included as needed to account for time changes in each county such as cultural economic conditions, energy transition policies and local infrastructure.

We reran the regression models. The results are shown in Table R8. From Columns (1) and (3), we can see that the gap in unplanned outages due to natural disasters between poverty and non-poverty counties does not change significantly when both are impacted simultaneously within the same city. However, the result in Column (2) shows the gap in planned outage frequency decreases significantly in these cases. This decrease may be related to power companies' scheduling decisions, as non-poverty counties may implement more planned outages during disasters to maintain grid safety, thereby reducing the difference in the impact of natural disasters on planned outages between poverty and non-poverty counties. Overall, our results may not validate the exacerbation of inequality between poverty and non-poverty counties during simultaneous natural disasters. We are grateful for the reviewers' insightful comments,

which have enriched our research findings.

Table R8: Impacts of grid-related natural disasters on power outages across counties during simultaneous outages

	ln (Outage frequency)		ln (Outage duration)	
	(1)	(2)	(3)	(4)
	Unplanned	Planned	Unplanned	Planned
Natural_disaster	-0.0192	-0.0423**	-0.0359	-0.0656
*Poverty*Both				
	(0.0153)	(0.0193)	(0.0322)	(0.0439)
Natural_disaster	0.0316***	0.0061**	0.0481***	0.0038
	(0.0023)	(0.0027)	(0.0038)	(0.0053)
Both	0.0239***	-0.0005	0.0385***	0.0109
	(0.0054)	(0.0080)	(0.0112)	(0.0186)
Natural_disaster*Poverty	0.0222**	0.0048	0.0536**	0.0105
	(0.0106)	(0.0128)	(0.0216)	(0.0291)
Natural_disaster*Both	-0.0185***	0.0187**	-0.0195	0.0363
	(0.0065)	(0.0094)	(0.0130)	(0.0222)
Both*Poverty	0.0085	0.0237*	0.0022	0.0340
	(0.0095)	(0.0132)	(0.0203)	(0.0293)
Control	YES	YES	YES	YES
County*YM	YES	YES	YES	YES
Observation	1497450	1497450	1504150	1504150
R2	0.9635	0.8628	0.0981	0.2563

Notes: *p<0.10, **p<0.05, *** p<0.01. Standard errors are clustered at the county level and reported below the coefficients. Natural disaster represents the intensity of natural disasters. Poverty is a dummy variable, equaling 1 if the county is a poverty county. Both is equal to 1 when both a poverty-stricken county and a non-poverty county within the same city experience a natural disaster simultaneously, otherwise 0. Control variables include dummy variables for holidays and weekends. County*YM is the

county-month-by-year fixed effect. Observation is the sample size. R2 represents the goodness-of-fit of the regressions.

Comments: 4. Regarding to power outage loss, it may not be the case that the higher frequency and longer duration power outages will have higher loss. We should also consider the load priority.

Response:

We thank the reviewer for the suggestion about the necessity of considering the load priority in the loss calculation. Following the reviewer’s suggestion, we refine our analysis by incorporating load priority into the assessment of power outage losses. We have recalculated the potential economic losses from power outages caused by natural disasters, assigning different loss values to counties based on the proportion of outages affecting various sectors in each county and the corresponding losses for each sector. We have addressed this issue through the following three aspects.

First, to capture a comprehensive impact of different natural disasters on power outages, we construct an indicator (*Total_outage_hours*), which is designed to simultaneously capture outage frequency and duration characteristics. *Total_outage_hours* equals per capita outage frequency (*Outage_frequency*) in the county on that day multiplied by the average duration per outage (*Outage_duration*).

Second, we estimate the impacts of different natural disasters on *Total outage hours* based on Equation (11), the results are shown in Table R9. Specifically, in poverty counties, the strong wind, rainstorms, cold waves and heatwave increase total outage hour by 0.0693, 0.9272, 0.0578 and 0.0206, respectively, compared to 0.0425, 0.1710, 0.0120 and 0.0127 in non-poverty counties.

$$Total_Outage_hour_{id} = \beta_1 Strong\ wind_{id} + \beta_2 Rainstorm_{id} + \beta_3 Cold\ wave_{id} + \beta_4 Geo\ hazard_{id} + \beta_5 Wildfire_{id} + \beta_6 Heatwave_{id} + X'\theta + \delta_{ym} + \varepsilon_{id} \quad (11)$$

In Equation (11), β_j represents the impact of j natural disaster on power outage. The variables are interpreted in the same manner as previously described.

Then we account for different types of loads (including residential, industrial commercial, and mixed-used loads) to project the economic losses caused by power outages in Equation (12).

$$Loss = \sum_i \sum_i \sum_l load_l * weight_{il} * \sum_j \beta_j * \Delta Nature\ disaster_{jt} * ele_h_i * ad \quad (12)$$

In Equation (12), $load_l$ represents the average economic loss per kilowatt-hour for types l of loads (residential, industrial commercial, and mixed-used loads). Based on the work of Zhang and Huang (2022), the average economic loss for the residential load, industrial load, commercial load, and mixed-used load is 13, 160, 109, and 112 RMB/kWh, respectively. $weight_{il}$ represents the proportion of each type of load in county i affected by outages. $\Delta Nature\ disaster_{jt}$ represents the change in the j-th type of natural disaster under the Shared Socioeconomic Pathways-Representative Concentration Pathways (SSP-RCP) scenarios from the year 2022 to 2100, relative to the period 2019-2021. The ele_h_i is the electricity consumption per hour for county i . ad is an average adjustment factor, and represents the number of people affected by each outage. Based on our power outage dataset, each outage affects approximately 1,330 households. The average household size is 2.6 people (data sourced from the National Bureau of Statistics of China).

Based on our calculations, the projected damages between 2022 and 2100 are estimated to range from 549 to 1340 billion RMB. The increase in future disasters is projected to widen the outage gaps caused by natural disasters between poverty and non-poverty counties under the SSP-RCP126 (157.82%) and SSP-RCP585 (157.76%) scenarios.

Table R9 Impacts of different grid-related natural disasters on total outage hours

	Total_outage_hours	
	(1)	(2)
	Poverty	Non_Poverty
Strong_wind	0.0693*** (0.0258)	0.0425*** (0.0144)
Rainstorm	0.9272*** (0.2966)	0.1710*** (0.0463)
Cold_wave	0.0587** (0.0271)	0.0120*** (0.0044)
Geo_hazard	0.0578 (0.0410)	0.0130 (0.0099)
Wildfire	0.1653 (0.1278)	-0.0153 (0.0103)
Heatwave	0.0206 (0.0159)	0.0127*** (0.0030)
Control	YES	YES
County*YM	YES	YES
Observation	306190	1191260
R2	0.0975	0.0805

Notes: *p<0.10, **p<0.05, *** p<0.01. Standard errors are clustered at the county level and reported below the coefficients. Control variables include dummy variables for holidays and weekends. County*YM is the county-month-by-year fixed effect. Observation is the sample size. R2 represents the goodness-of-fit of the regressions.

We would like to express our gratitude once again to the reviewer for their valuable suggestions. In the revised manuscript, these details have been added to the Results Section (Page 10, lines 42-44; Page 11, lines 1-7).

The projected increase in natural disasters is expected to result in 0.21 h/person/year

from 2022 to 2100 under the SSP-RCP126 scenario, and 0.41 h/person/year under the SSP-RCP585 scenario. We also perform a back-of-the-envelope calculation of the economic cost of power outage hours induced by natural disasters (see Methods). Based on our calculations, the projected damages from 2022 to 2100 are estimated to range from 549 to 1340 billion RMB. We find that climate change will further widen the outage gaps caused by natural disasters between poverty and non-poverty counties in SSP-RCP126 (157.82%) and SSP-RCP585 scenarios (157.76%). These findings indicate that the reliability of power supply in poverty counties will face substantial challenges in the future.

Reference

Zhang, W., and Huang, Z. (2022). Research on priority strategy of overhaul without power interruption based on user outage loss assessment (in Chinese). *Electric Engineering 01*, DOI:10.19768/j.cnki.dgjs.

Comments: Reviewer #2 (Remarks on code availability): The reviewer thinks that the data availability regarding the high-resolution power outage data may be an issue for others to reproduce the analysis.

Response:

We thank the reviewer for this important comment. To address this concern, we have further justified the availability regarding the high-resolution power outage data from the following two aspects and provided the steps of reproduction in Data availability and Code availability Section.

From a regulatory requirement perspective, according to the "Electricity Supply Supervision Measures" (State Electricity Regulatory Commission Order No. 27), users need to be notified of power outages due to the planning maintenance and temporary maintenance of power supply facilities. Similarly, 'Electric Power Enterprise Information Disclosure Regulations' mandate that power enterprises disclose relevant information, including power outage notifications. These regulations are designed to protect consumers' right to know, ensure the transparency of electricity supply, and provide users with ample time to respond to power outages.

From the practical perspective of data accessibility, the local public utility websites will provide real-time power outage information and estimated recovery time, including planned outages and unplanned outages. We could obtain high-resolution power outage information, such as the start and end time, addresses and areas affected, and causes of power outages, on public utility websites, such as <https://95598.impc.com.cn/infomation/tdtz>. Third-party websites of <http://www.sttcq.com/td/> compile power outage information across China. The information in Figure R1 displays a screenshot of a power outage record from Huoqiu County, Liuan City. By tracking and scraping real-time outage information from these websites, we were able to construct the outage indicators used in this study.

[Figure Redacted]

(b)

(b)

Figure R1. A screenshot of power outage record for Huoqiu County, Liuan City, China

Note: Panel (a) illustrates the original power outage record in Chinese for Huoqiu county, Liuan City, China.

Panel (b) presents the translation power outage record in English for the same location.

We also provided the data and code on <https://github.com/Adaptability1/outage.git>. named “data_natural.dta” for the reproduction of results. Readers could use the Stata application to load the “data_natural.dta” file, then run the “main_do.do” file to get the results. The “.do” and “.dta” files are the code and data for the main regression results in the manuscript. We also offer more detailed instructions for “.do” and “.dta” files in the README section on the website <https://github.com/Adaptability1/outage.git>.

We once again appreciate the valuable suggestion and hope our response has adequately addressed the reviewer’s concerns. In the revised manuscript, we have added the

description of data availability in the Method Section (Page 11, lines 14-24).

The high-resolution power outage data in China were collected from local public utility websites. Given the significant impact of power outages on both residential and industrial/commercial activities, the power outage data is made publicly available (though they are not being systematically collected and stored in a central database), and the users can get the real-time power outage information from the local public utility websites. We aggregated power outage data at the county and daily levels, covering the period from November 2019 to September 2021. The outage data includes the location of the power outage, starting time, ending time and type of power outage (planned or unplanned). Here, planned outages are notified in advance. Unplanned power outages are unexpected power failures, that may be caused by natural and man-made factors (Babar et al., 2022; Waseem and Manshadi, 2020).

Reviewers comments:

Reviewer #1 (Remarks to the Author):

Comments: Thank you for addressing my comments point-by-point. In general, most of my concerns have been addressed to some extent, though some still require further discussion and analysis in the revised manuscript. The following comments aim to make the assumptions more practical and strengthen the conclusions:

Response:

Thank you for your remarks on the overall work. We are grateful for your constructive feedback and the opportunity to further improve our work.

Comments: 1. For the outage loss estimation in the Methods section, the use of population-based normalization might not be appropriate for this study, as the primary focus is to analyze the relationship between natural disasters and economic development levels to power outages. Please conduct a comparative analysis, using GDP (either total or per capita) as the basis for normalization.

Response:

Thank you for your insightful comment. In response to your suggestion, building upon the original population-based analysis, we have now added a total GDP-based analysis to serve as a comparison. The updated estimation process now includes three key aspects: First, to analyze the relationship between natural disasters and economic development levels to power outages, we construct an indicator, *Total_outage_hours2*. This indicator is calculated by multiplying the regional total GDP-adjusted outage frequency (*Outage_frequency2*) in a county on a given day by the average outage duration (*Outage_duration*). As an example, we use the power outage situation in Pianguan County, Shanxi, on October 15th, 2021, to illustrate the calculation steps for *Outage_frequency2*, *Outage_duration* and *Total_outage_hours2*. Pianguan County experienced two power outages on that day: the first one lasting 2.75 hours, and the second one lasting 7.5 hours. The total GDP of Pianguan County is 3.29251 billion RMB. The *Outage_frequency2* (GDP-adjusted outage frequency) is calculated to be 0.60744 ($2/3.29251$). And the *Outage_duration* is calculated to be 5.125

$((2.75+7.5)/2)$ hours. The $Total_outage_hours2$ equals to $3.11(2/3.29251*(2.75+7.5)/2)$. This approach ensures that both the frequency and severity of power outages are considered in a manner that reflects regional economic development levels.

Second, we employ Equation (1) to estimate the marginal impact of natural disasters on $Total_outage_hours2$ in poverty and non-poverty counties.

$$Total_Outage_hours2_{id} = \beta_1 Strong_wind_{id} + \beta_2 Rainstorm_{id} + \beta_3 Cold_wave_{id} + \beta_4 Geo_hazard_{id} + \beta_5 Wildfire_{id} + \beta_6 Heatwave_{id} + X'\theta + \delta_{iym} + \varepsilon_{id} \quad (1)$$

In Equation (1), β_j represents the impact of j-th natural disaster on a power outage. $Strong_wind_{id}$ ($Rainstorm_{id}$, $Cold_wave_{id}$, Geo_hazard_{id} , $Wildfire_{id}$ and $Heatwave_{id}$) is a dummy variable indicating whether a county experiences strong winds (rainstorms, cold waves, geological disasters, wildfire and heatwaves) on day d in county i and 0 otherwise. The covariates X' control the holiday and weekend dummy variables. The county-month-by-year fixed effects (δ_{iym}) are included as needed to account for time changes in each county such as cultural and economic conditions, energy transition policies, and local infrastructure. We cluster the standard errors at the county level which is consistent with the sampling of our data.

The results are shown in Table R1. Specifically, in poverty counties, the strong wind, rainstorms, cold waves and heatwave increase total outage hours by 0.0271, 0.2514, 0.0163 and 0.0027, respectively, compared to 0.0108, 0.0291, 0.0025 and 0.0018 in non-poverty counties.

Table R1. Impacts of grid-related natural disasters on total power outage hours based on the GDP-based normalization

	Total_outage_hours2	
	(1)	(2)
	Poverty	Non_Poverty
Strong_wind	0.0271*** (0.0102)	0.0108*** (0.0040)
Rainstorm	0.2514*** (0.0741)	0.0291*** (0.0072)
Cold_wave	0.0163** (0.0071)	0.0025** (0.0011)
Geo_hazard	0.0153 (0.0106)	0.0026 (0.0022)
Wildfire	0.0654 (0.0530)	-0.0013 (0.0022)
Heatwave	0.0027 (0.0042)	0.0018*** (0.0006)
Control	YES	YES
County*YM	YES	YES
Observation	304180	1030460
R ²	0.0938	0.1096

Note: The statistical significance of the coefficients is indicated by asterisks: *** p < 0.01, ** p < 0.05, * p < 0.10. Standard errors are clustered at the county level and are reported in parentheses. County-by-month-by-year fixed effects (County*YM) are included. The number of observations and R² are reported at the bottom of the table.

Then, we account for different types of loads (including residential, industrial, commercial, and mixed-used loads) to project the economic losses caused by power outages in Equation (2). We focus on the change of power outages caused by strong winds, heatwaves, cold waves, and rainstorms. The details of the Equation are as follows:

$$Loss = \sum_i \sum_i \sum_l load_l * weight_{il} * \sum_j \beta_j * \Delta Nature\ disaster_{jt} * ele_h_i * ad2 \quad (2)$$

In Equation (2), $load_l$ represents the average economic loss per kilowatt-hour for the

different types of loads (residential, industrial, commercial, and mixed-used loads). Based on the work of Zhang and Huang (2022), the average economic loss for the residential load, industrial load, commercial load, and mixed-used load is 13, 160, 109, and 112 RMB/kWh, respectively. $weight_{it}$ represents the proportion of each type of load in county i affected by outages. $\Delta Nature\ disaster_{jt}$ represents the change in the j -th type of natural disaster under the Shared Socioeconomic Pathways-Representative Concentration Pathways (SSP-RCP) scenarios from the year 2022 to 2099, relative to the period 2019-2021. The ele_h_i is the electricity consumption per hour for county i . $ad2$ is an average adjustment factor, and represents the magnitude of GDP affected by each outage.

Based on our calculations, the projected damages between 2022 and 2099 are estimated to range from 257 to 481 billion RMB, which is slightly higher than the losses calculated using population-normalized results (192 to 468 billion RMB). The details of these adjustments have been incorporated into the Results Section of the revised manuscript (Page 12, lines 3-5).

Comments: 2. In the Results section, Beyond the general principle that "natural disasters induce more power outages," are there countries where fewer outages occur under similar natural disaster conditions? These counterintuitive observations could provide valuable insights and help decision-makers assess the robustness of the derived conclusions.

Response:

Thanks for this insightful comment. Follow the reviewer's suggestion, we identify counties where the total frequency of natural disasters ranked within the top 10%, and the total frequency of power outages ranked within the bottom 10% during the study period. These counties account for only 0.67% of the total counties. We further find that all of these counties are non-poverty counties. Given that our research focus is on the differences between poverty and non-poverty counties, we do not further investigate

the underlying causes of this phenomenon "more natural disasters and less power outages" for these counties, as they do not align with our research priorities. If appropriate data are available, we will further explore the broader causes in future research.

Additionally, inspired by your comments, we try to provide more heterogeneity analysis to offer more valuable insights. In order to compare the outage under similar natural disaster conditions, we categorize the cumulative number of natural disasters during the study period into two groups: 0-10 occurrences and more than 10 occurrences during the study period, and take poverty counties and non-poverty counties as the main heterogeneity analysis objects. We compare the severity of the power outages between poverty counties and non-poverty counties under similar natural disaster conditions. Additionally, since the study focuses on the impact of power outages caused by natural disasters, we also compare the impact of natural disasters on power outages between poverty counties and non-poverty counties under similar natural disaster conditions.

Firstly, we show the severity of the power outages in both poverty and non-poverty counties under similar natural disaster conditions, and then compare their differences using a T-test. The results are shown in Table R2. We can see that, when natural disasters become more frequent, power outages will significantly increase in both poverty and non-poverty counties (see results in Lines 3 and 6). And focusing on the research subjects and conclusions of this paper, we further analyze the differences between poverty and non-poverty counties. We find that households in poverty counties experience both more frequent power outages and longer power outage duration per occurrence, regardless of whether the regions are exposed to more than 10 natural disasters or 0-10 natural disasters during the study period(see results in Columns 3 and 6).

Table R2 Between-group difference T-test of power outage

		Non_Poverty Counties	Poverty Counties	T-test (Poverty Counties - Non-Poverty Counties)
Outage frequency	0~10 occurrence	0.0052	0.0086	0.0034***
	>10 occurrence	0.0089	0.0101	0.0012***
T-test (>10 - 0~10 occurrence)		0.0037 ***	0.0015***	
Outage duration	0~10 occurrence	1.0538	1.5119	0.4581***
	>10 occurrence	1.8312	2.6277	0.7965***
T-test (>10 - 0~10 occurrence)		0.7774***	1.1158***	

Note: The statistical significance of the T-test is indicated by asterisks: *** $p < 0.01$, ** $p < 0.05$, * $p < 0.10$.

Secondly, we compare the impact of natural disasters on power outages between poverty counties and non-poverty counties under similar natural disaster conditions. The results show in Table R3 and Table R4. The results show that the impact of natural disasters on power outages is significantly greater in poverty counties compared to non-poverty counties when the regions experience 0-10 natural disasters in our study period.

There is no significant difference in the impact between poverty and non-poverty counties when the regions experience more than 10 natural disasters. This phenomenon may be attributed to several underlying reasons. The repeated natural disasters often cause extensive and severe damage to power infrastructure, leading to prolonged and frequent power disruptions for both poverty and non-poverty counties (Masozera et al, 2006; Chester et al, 2020). Furthermore, in areas frequently hit by natural disasters, both

poverty and non-poverty counties often receive substantial external assistance for post-disaster power grid recovery, including financial resources, human resources, and technical support, which makes the impact of inherent economic disparities on outage severity less pronounced (Brullo et al, 2024). Due to the limitations of data, we are currently unable to find sufficient empirical evidence to explain the underlying true causes. However, this does not affect our main conclusions. We will explore the underlying mechanism in future studies.

Table R3. Impacts of grid-related natural disasters on power outage frequency in the counties with varying frequencies of natural disasters

	ln(Outage frequency)			
	0~10 occurrence during the study period		>10 occurrence during the study period	
	(1)	(2)	(3)	(4)
Natural disaster	0.0165*** (0.0047)	0.0055 (0.0048)	0.0443*** (0.0033)	0.0431*** (0.0035)
Natural disaster*Poverty		0.0469*** (0.0129)		0.0084 (0.0102)
Control	YES	YES	YES	YES
County*YM	YES	YES	YES	YES
Observation	644540	644540	852910	852910
R ²	0.8992	0.8992	0.7883	0.7883

Notes: The statistical significance of coefficients is indicated by asterisks: *** p < 0.01, ** p < 0.05, * p < 0.10. Standard errors are clustered at the county level and reported in parentheses. County*YM is the county-month-by-year fixed effect.

Table R4. Impacts of grid-related natural disasters on power outage duration in the counties with varying frequencies of natural disasters

	ln(Outage duration)			
	0~10 occurrence during the study period		>10 occurrence during the study period	
	(1)	(2)	(3)	(4)
Natural disaster	0.0313*** (0.0095)	0.0140 (0.0102)	0.0635*** (0.0064)	0.0592*** (0.0066)
Natural disaster*Poverty		0.0742*** (0.0251)		0.0311 (0.0213)
Control	YES	YES	YES	YES
County*YM	YES	YES	YES	YES
Observation	647890	647890	856260	856260
R ²	0.2059	0.2059	0.2615	0.2615

Notes: The statistical significance of coefficients is indicated by asterisks: *** $p < 0.01$, ** $p < 0.05$, * $p < 0.10$. Standard errors are clustered at the county level and reported in parentheses. County*YM is the county-month-by-year fixed effect.

Once again, we appreciate the valuable suggestion and hope our response has adequately addressed your concerns. In the revised manuscript, we have added the details in the Results Section (Page 7, lines20-31; Page 10, lines17-22).

Reference

Masozera M , Bailey M , Kerchner C .Distribution of Impacts of Natural Disasters Across Income Groups: A Case Study of New Orleans[J].Ecological Economics, 2007, 63(2-3):299-306.DOI:10.1016/j.ecolecon.2006.06.013

Chester, M., Underwood, B.S. & Samaras, C. Keeping infrastructure reliable under climate uncertainty. Nat. Clim. Chang. 10, 488–490 (2020).
<https://doi.org/10.1038/s41558-020-0741-0>

Brullo, T., Barnett, J., Waters, E. et al. The enablers of adaptation: A systematic review. *npj Clim. Action* 3, 40 (2024). <https://doi.org/10.1038/s44168-024-00128-y>

Comments: 3. In the Subsection “Power outage and grid natural disaster”, The relationship between natural disasters and power outages warrants further investigation, especially since power outages are not always caused by natural disasters. For example, planned outages are often published by public utility websites. It would be helpful to clarify and justify why power outages in this study are assumed to be primarily induced by natural disasters.

Response:

Thanks for this insightful comment. We appreciate your suggestion to further clarify the relationship between natural disasters and power outages, particularly since power outages are not always caused by natural disasters. In response, we would like to provide additional context and justification for our focus on natural disasters. We ensure our study focuses on power outages induced by natural disasters in the following two ways:

On one hand, according to the *2021 China Power Reliability Annual Report*, approximately 27.33% of the power outages per household in China are caused by natural factors. Given that natural factors are a critical determinant of power outages and are expected to have an increasing impact due to future climate change, this study focuses on the impact of natural disasters on power outages. Our study explores the causal effect of natural disasters on power outages by controlling for confounding factors using econometric models. Specifically, we introduce county-month-by-year fixed effects in the fixed effect models. The county-month-by-year fixed effects are used to capture the monthly impacts of economic conditions, energy transition policies and local infrastructure across different years at the county level on power outages. By including the county-month-by-year fixed effects, we can eliminate the influence of these unobservable factors, making the estimation more reflective of the isolated effect

of natural disasters on power outages.

On the other hand, we further distinguish between planned and unplanned power outages in our research. Planned outages are pre-scheduled and notified in advance, while unplanned outages are unexpected and can be triggered by both natural and human-induced factors. In the revised manuscript, we further compare the impact of natural disasters on planned and unplanned outages. The results are shown in Table R5. We find that natural disasters lead to a 3.50% increase in the frequency of unplanned outages and a 1.04% increase in planned outages. Natural disasters extend the duration of unplanned outages by 5.78% and planned outages by 1.65%. Although natural disasters also contribute to an increase in planned outages, this is likely due to heightened maintenance efforts needed after the disasters that are aimed at stabilizing the power grid and mitigating potential risks.

Table R5. Impact of natural disaster on unplanned and unplanned outages.

	ln(Outage frequency)		ln(Outage duration)	
	Unplanned	Planned	Unplanned	Planned
Natural disaster	0.0350*** (0.0018)	0.0104*** (0.0023)	0.0578*** (0.0033)	0.0165*** (0.0049)
Control	YES	YES	YES	YES
County*YM	YES	YES	YES	YES
Observation	1497450	1497450	1504150	1504150
R ²	0.9635	0.8628	0.0980	0.2563

Notes: *p<0.10, **p<0.05, *** p<0.01. Standard errors are clustered at the county level are reported in parentheses. County-by-month-by-year fixed effects (County*YM) are included. The number of observations and R² are reported at the bottom of the table.

Furthermore, to assess the relative impact of natural disasters on planned versus unplanned outages, we conduct a T-test comparing the coefficients of the two types of outages (see Table R6). The results show a significant difference, indicating that unplanned outages are far more strongly impacted by natural disasters than planned outages.

Table R6. Between-group Difference T-test

		Unplanned	Planned	T-statistic (Unplanned-Planned)
Natural disaster	ln(Outage frequency)	0.0350*** (0.0018)	0.0104*** (0.0023)	0.0246*** (8.4534)
	ln(Outage duration)	0.0578*** (0.0033)	0.0165*** (0.0049)	0.0413*** (7.0409)

Note: The statistical significance of the T-test is indicated by asterisks: *** $p < 0.01$, ** $p < 0.05$, * $p < 0.10$. The numbers in parentheses in the first two columns represent standard errors. The numbers in parentheses in the last column reports the t-values.

Once again, we appreciate the valuable suggestion and hope our response has adequately addressed your concerns. In the revised manuscript, we have added the details in the Results Section (Page 7, lines7-18).

Comments: 4. Regarding the country-level investment information, please provide the data sources and, if possible, make them open-source. This data directly impacts the conclusions in the “Power Outages Caused by Natural Disasters in Poverty-Stricken Counties” subsection. Additionally, national-level investments should also be considered, as they can influence the impacts and losses, either directly or indirectly.

Response:

Thanks for your insightful comment. We appreciate your suggestion regarding providing data sources for national-level investment information and exploring its impact in the analysis. In response, we have made the power grid investment data publicly available on <https://github.com/Adaptability1/outage.git>, which includes data from the State Grid Corporation of China and their respective provincial subsidiaries. This data is sourced from the 'Compilation of Electric Power Industry Statistical Data'.

Following your advice, we have incorporated the impact of grid national-level

investments and region-level investments on power outages caused by natural disasters. According to 'Compilation of Electric Power Industry Statistical Data', the investment and construction of the power grid in China are primarily undertaken by the provincial power grid companies' subsidiaries of State Grid Corporation of China, accounting for 95.04% of the total investment from 2016 to 2019. The remaining 4.96% is invested by the headquarters of State Grid Corporation of China.

Based on this data, we examine the impact of grid investments (including the national-level and region-level grid investments) on power outages caused by natural disasters by introducing interaction terms of natural disasters and grid investments (Natural disaster*EGI). The updated results are shown in Table R7. We find that increased investment in grid infrastructure can significantly mitigate the frequency and duration of power outages during natural disasters.

Table R7. Impact of grid investment on natural-disaster-induced outages.

	(1)	(2)
	ln(Outage frequency)	ln(Outage duration)
Natural disaster*EGI	-0.1594*** (0.0436)	-0.4338*** (0.0784)
Natural disaster	0.0537*** (0.0049)	0.0956*** (0.0096)
Control	YES	YES
County*YM	YES	YES
Observation	1497450	1504150
R2	0.8410	0.2505

Note: *p<0.10, **p<0.05, *** p<0.01. Standard errors are clustered at the county level are reported in parentheses. County*YM is the county-month-by-year fixed effect. R² indicates the goodness-of-fit of the regressions.

Once again, we appreciate the valuable suggestion and hope our response has adequately addressed your concerns. In the revised manuscript, we have added the details in the Results Section (Page 8, lines 35-42).

Comments: 5. In Subsection “Power outages caused by natural disasters in poverty counties”, using the Shapley value to decompose the impact of disasters on power outages should be further validated, particularly since rainstorms and strong winds are often highly correlated during typhoon events. The Shapley value assumes that contributions can be fairly separated, but these disasters typically occur in a dependent or sequential pattern, which may complicate the application of this method.

Response:

Thank you for your insightful comments regarding the correlation between rainstorms and strong winds. We first investigate the probability of the co-occurrence of rainstorms and strong winds. As the reviewer pointed out, rains and winds are often correlated. The average rainfall accompanying strong winds is 3.3 mm during the strong wind days, and the average wind speed accompanying rainstorms is 5.45 m/s during the rainstorm days. However, this study is mainly concerned with natural disasters rather than common winds and rains. According to the criteria of this study, strong winds are defined as a day’s maximum wind speeds above 10.8 m/s, and rainstorms are defined as precipitation of more than 50 mm in a single day. The probability of these two extreme weather events occurring simultaneously on days with either rainstorms or strong winds is merely 0.67%. The number of cases where both reach the thresholds set in this paper for defining a disaster is not large.

Secondly, to mitigate the impact of this correlation on our Shapley decomposition, we followed reviewer advise and have refined our approach by separating rainstorms and strong winds into three distinct variables: (1) rainstorms occurring alone, (2) strong winds occurring alone, and (3) the simultaneous occurrence of rainstorms and strong winds. By using the three variables separately, we reduce the potential correlation between these events and allow for a clearer decomposition of their individual contributions to power outages. Then, we use the Shapley value method to reevaluate the separated contribution of different natural disasters to power outages.

Based on the Shapley value approach and the subsamples we have set above, we find that 44.16% of the impact of disasters on power outage frequency is attributed to rainstorms occurring alone, 7.62% is attributed to strong winds occurring alone, 0.01% is attributed to the simultaneous occurrence of rainstorms and strong winds (see in Table R8). Similarly, for power outage duration, 43.39% of the impact is attributed to rainstorms occurring alone, 7.85% is attributed to strong winds occurring alone, and only 0.11% is attributed to the simultaneous occurrence of rainstorms and strong winds. As we mentioned above, although rain and winds are often highly correlated, the probability of both reaching disaster thresholds on the same day is only 0.67%. In most cases, when wind speed reaches the threshold, the average rainfall is 3.3 mm, which is still far from reaching the rainstorm disaster threshold. When heavy rain reaches the threshold, the average wind speed is only 5.45 m/s, which is still far from reaching the strong wind disaster threshold. Our analysis in Table R8 indicates poverty counties are more significantly affected by rainstorms, strong winds, and the simultaneous occurrence of rainstorms and strong winds than non-poverty regions.

Table R8. The decomposition results of natural disaster impact on power outages based on Shapely Values considering the correlation of strong winds and rainstorm

	ln(Outage frequency)			ln(Outage duration)		
	All	Povert y	Non- poverty	All	Povert y	Non- poverty
Strong_wind_alone	7.62%	18.21 %	5.10%	7.85%	9.00%	7.38%
Rainstorm_alone	44.16 %	50.65 %	41.93%	43.39 %	49.77 %	42.04%
Strong_wind_Rainstor m	0.01%	0.44%	0.04%	0.11%	0.91%	0.05%
Cold_wave	10.99 %	12.98 %	9.74%	15.47 %	14.58 %	14.99%
Geo_hazard	5.88%	8.81%	4.50%	4.82%	11.49%	2.33%

Wildfire	0.04%	0.06%	0.03%	0.87%	0.99%	0.71%
Heatwave	31.30%	8.85%	38.66%	27.49%	13.27%	32.49%

Once again, we appreciate the valuable suggestion and hope our response has adequately addressed your concerns. We have updated these details in our revised manuscript, which can be found in the Results Section (Page 10, lines 36-38; Page 11, lines 6-8).

Comments: 6. Reviewer #1 (Remarks on code availability): Natural disaster and power outage information are published, while the country-level investment information is missing. The correlation among natural disasters should be analyzed when the Sharly value is adopted.

Response:

Thank you for your valuable comment. We have provided the power grid investment data and further reevaluated the contribution of different natural disasters on power outages based on Shapely values considering the correlation among natural disasters.

First, we provide the power grid investment data for the State Grid Corporation of China, and the respective provincial subsidiaries on <https://github.com/Adaptability1/outage.git>, all of which are sourced from the 'Compilation of Electric Power Industry Statistical Data'.

Second, to address the issue of correlation among natural disasters, we investigate the probability of the co-occurrence of rainstorms and strong winds. Although rains and winds are often correlated, this study is mainly concerned with natural disasters level winds and rains. The probability of these two extreme weather events occurring simultaneously on days with either rainstorms or strong winds is merely 0.67%. The number of cases where both reach the thresholds set in this study for defining a disaster

is not large. To address the potential issue of correlation between natural disasters, particularly the interplay between rainstorms and strong winds, we refined our Shapley decomposition method to ensure a more robust and accurate attribution of their individual and interactive impacts. For the specific reply, please refer to the response to comment 5. We have also adjusted the code, which has been updated on <https://github.com/Adaptability1/outage.git>.

Finally, we once again extend our sincere thanks to the reviewer for the insightful comments and evaluations, which have played a significant role in enhancing the precision and depth of our research.

Reviewers comments:

Reviewer #1 (Remarks to the Author):

Comments: Thank you for your point-by-point responses to my previous comments. Some issues still require further discussion and analysis in the revised manuscript:

Response:

Thank you for your remarks on the overall work. We are grateful for your constructive feedback and the opportunity to improve our work further.

Comments: 1. The introduction lacks proper citations for certain data points, such as the percentage of power outages caused by natural factors in China. While you mentioned the "2021 China Power Reliability Annual Report" in your response, to my knowledge, this report doesn't clearly define what constitutes "natural disasters." In practice, outages caused by extreme weather events are typically categorized and counted separately as major incidents rather than being included in routine reliability statistics for distribution networks. The "natural factors" in reliability statistics usually refer to more common occurrences like tree branch contacts. Could you please further discuss the correlation between the outage dataset and natural disaster dataset used in your study?

Response:

Thank you for this thoughtful and constructive comment. In response, we have removed the sentence from the introduction that previously referenced the percentage of power outages caused by natural factors, in order to avoid potential misrepresentation due to inappropriate or ambiguous citation.

Additionally, we have added a clearer explanation of the relationship between the natural disaster dataset and the power outage dataset used in our study. Specifically, these two datasets are independent: the power outage data is sourced from local public utility websites, which release real-time information on power outage events for public notification and transparency, while the county-level natural disaster data comes from

the National Cryosphere Desert Data Center (NCDC) and the Meteorological Data Service Centre of China (MDSCC). We matched the power outage data and natural disaster data based on the longitude and latitude information of the counties.

Although the power outage dataset does not explicitly document the cause of each outage, we can employ econometric analysis to estimate the causal relationship between natural disaster events and power outage outcomes. Specifically, we utilize a high-dimensional fixed-effects model to estimate the marginal effect of natural disasters on outage frequency and duration, which allows us to isolate and quantify how each additional unit increase in disaster intensity affects power outages. The specific estimation method can be found on Page 16 (lines 16-27), Page 17 (lines 1-32) and Page 18 (lines 1-3) of the revised manuscript.

Comments: 2. The outage examples provided in the introduction (e.g., cold waves and Sichuan-Chongqing power restrictions) don't strongly relate to your core theme. These were large-scale events that significantly affected both urban and non-poor areas. I recommend focusing more on examples that highlight the unique characteristics and challenges of power outages in impoverished regions. Additionally, your result figures show a notable spatial discrepancy between disaster distributions and poverty-stricken areas, which seems counterintuitive to your main conclusions.

Response:

We appreciate your insightful feedback regarding the examples provided in the introduction. To address this, we have revised the introduction to include examples that more accurately reflect the unique vulnerabilities of impoverished communities during power outages. The relevant content appears in our revised manuscript on Page 2, Lines 4-10.

In May 2019, severe rainfall and flooding in Guangxi's Guilin and Chongzuo regions led to widespread power outages across rural communities. In June 2020, a landslide in Aniangzhai Village, Danba County, Sichuan Province, caused two 10-kilovolt lines to go out, and interrupted

electricity supply to 4,507 rural households. These disruptions significantly impacted household daily life, underscoring the limited disaster resilience of vulnerable regions compared to urban areas.

Regarding the spatial discrepancy between disaster and poverty areas in Figure 1, we clarify that the figure primarily visualizes the raw geographic distributions of power outages and natural disasters across China. Importantly, our core argument is not that poverty counties face higher disaster exposure than non-poverty counties, but rather that they experience a greater marginal increase in power outages during natural disasters based on the econometric modeling analysis.

Specifically, we employ a fixed-effects model by controlling for county-month-by-year fixed effects and other observable covariates. This empirical strategy allows us to isolate the marginal impact of natural disasters on power outage outcomes in both poverty and non-poverty counties. Table R1 shows the main results. The regression results support our main conclusion: the poverty counties experience a higher frequency and longer duration of power outages caused by natural disasters compared to non-poverty counties. Specifically, natural disasters increase the outage frequency by 5.19% for poverty counties, and by 3.80% for non-poverty counties (Columns 1 and 2, Table R1). Natural disasters extend the outage duration, resulting in an average increase of 8.96% for poverty counties and 5.34% for non-poverty counties (Columns 4 and 5, Table R1). We also introduce the interaction terms of natural disasters and the dummy variable of poverty counties to show the robustness of the results (Columns 3 and 6, Table R1). The results also show a higher impact of natural disasters on power outages in poverty counties than in non-poverty counties.

Table R1. Impacts of natural disasters on power outages in poverty and non-poverty counties

	ln(Outage frequency)			ln(Outage duration)		
	Poverty	Non-poverty	All	Poverty	Non-poverty	All
	(1)	(2)	(3)	(4)	(5)	(6)
Natural disaster	0.0519*** (0.0078)	0.0380*** (0.0031)	0.0381*** (0.0031)	0.0896*** (0.0161)	0.0534*** (0.0058)	0.0535*** (0.0058)
Natural disaster*Poverty			0.0143* (0.0083)			0.0373** (0.0171)
Control	YES	YES	YES	YES	YES	YES
County*YM	YES	YES	YES	YES	YES	YES
Observation	306190	1191260	1497450	306190	1197960	1504150
R ²	0.8464	0.8309	0.8410	0.2140	0.2615	0.2505

Notes: *p<0.10, **p<0.05, *** p<0.01. Standard errors are clustered at the county level and reported below the coefficients. Natural disaster represents the intensity of natural disasters. Poverty is a dummy variable, equaling to 1 if the county is a poverty county. Poverty2 is an alternative definition of poverty counties, valuing 1 if the county falls in the bottom third of per capita disposable income. Control variables include dummy variables for holidays and weekends. County*YM is the county-month-by-year fixed effect. Observation is the sample size. R² represents the goodness-of-fit of the regressions.

Comments: 3. The statement (lines 66-69) suggesting limited research on natural disasters' impact on power infrastructure resilience is inaccurate. This field has actually been studied for decades with substantial scholarly work. Please revise this section to provide a more comprehensive and precise discussion of existing research and its limitations.

Response:

Thank you for your valuable feedback. We have revised this section and provided a more comprehensive and precise discussion of existing research and its limitations according to your suggestions. Please refer to Page 2 (lines 15-44) and Page 3 (lines 1-3) of our revised manuscript for the updated content.

Despite the increasing frequency of natural disasters, there is a gap in empirical understanding of the impacts and corresponding implications on electric reliability in developing countries, particularly regarding comparative analyses across disaster types. The existing literature provides substantial insights into the impacts of natural disasters on the technical and infrastructural

resilience of power systems (Espinoza et al., 2016; Wang et al., 2016). Natural disasters such as hurricanes, wildfires, and floods could reduce the electricity supply by destroying the power infrastructure or damaging transmission and distribution lines (Alemazkoor et al., 2020; Casey et al., 2020; Mukherjee et al., 2018). However, the existing literature lacks systematic comparisons of how different natural disasters affect both the duration and frequency of power outages. Furthermore, predictive assessments of power outages and their associated economic loss under future climate change scenarios are limited. Addressing these gaps is crucial for developing targeted adaptation strategies and strengthening regional energy resilience.

In addition, the varying magnitudes of the impacts of natural disasters on power outages between poverty and non-poverty regions remain unclear. Power outages resulting from natural disasters may disproportionately affect impoverished individuals (Do et al., 2023). Poor people often live in regions with vulnerable public infrastructure, which is more susceptible to natural disasters (Soergel et al., 2021). In addition, poverty regions often lack backup generators during blackouts, which not only hampers the ability of vulnerable groups with chronic illnesses to seek medical services (Barreca et al., 2022; Hernández, 2016; Molinari et al., 2017) but also increases the risk of acute food shortages, as residents are unable to preserve food without refrigeration (Hecht et al., 2018). Therefore, the power outages caused by natural disasters and extreme weather may have unequal impacts on poor regions. Moreover, there is an evident empirical gap in our understanding of the mechanism for this inequality and whether this inequality will widen under plausible scenarios of climate change. The Sustainable Development Goals (SDGs) call for action to ensure access to affordable, reliable, sustainable, and modern energy for all (Fuso Nerini et al., 2018). Therefore, this study aims to provide empirical evidence on the inequitable and heterogeneous effects of natural disasters on electricity outages, thus highlighting the importance of energy justice in infrastructure investments.

References:

- [1] Espinoza S, Panteli M, Mancarella P, Rudnick H. Multi-phase assessment and adaptation of power systems resilience to natural hazards. *Electric Power Systems Research*. 2016;136:352-61.
- [2] Wang Y, Chen C, Wang J, Baldick R. *Research on Resilience of Power Systems Under Natural*

- Disasters—A Review. IEEE Transactions on Power Systems. 2016;31(2):1604-13.*
- [3] Mukherjee S, Nateghi R, Hastak M. A multi-hazard approach to assess severe weather-induced major power outage risks in the U.S. *Reliability Engineering & System Safety. 2018;175:283-305.*
- [4] Casey JA, Fukurai M, Hernández D, Balsari S, Kiang MV. Power Outages and Community Health: a Narrative Review. *Current Environmental Health Reports. 2020;7(4):371-83.*
- [5] Alemazkooor N, Rachunok B, Chavas DR, Staid A, Louhghalam A, Nateghi R, et al. Hurricane-induced power outage risk under climate change is primarily driven by the uncertainty in projections of future hurricane frequency. *Scientific Reports. 2020;10(1):15270.*
- [6] Do V, McBrien H, Flores NM, Northrop AJ, Schlegelmilch J, Kiang MV, et al. Spatiotemporal distribution of power outages with climate events and social vulnerability in the USA. *Nature communications. 2023;14(1):2470.*
- [7] Soergel B, Kriegler E, Bodirsky BL, Bauer N, Leimbach M, Popp A. Combining ambitious climate policies with efforts to eradicate poverty. *Nature Communications. 2021;12(1):2342.*
- [8] Barreca A, Park RJ, Stainier P. High temperatures and electricity disconnections for low-income homes in California. *Nature Energy. 2022;7(11):1052-64.*
- [9] Hernández D. Understanding 'energy insecurity' and why it matters to health. *Social science & medicine. 2016;167:1-10.*
- [10] Molinari NAM, Chen B, Krishna N, Morris T. Who's at risk when the power goes out? The at-home electricity-dependent population in the United States, 2012. *Journal of public health management and practice. 2017;23(2):152-9.*
- [11] Hecht AA, Biehl E, Buzogany S, Neff RA. Using a trauma-informed policy approach to create a resilient urban food system. *Public health nutrition. 2018;21(10):1961-70.*
- [12] Fuso Nerini F, Tomei J, To LS, Bisaga I, Parikh P, Black M, et al. Mapping synergies and trade-offs between energy and the Sustainable Development Goals. *Nature Energy. 2018;3(1):10-5.*

Comments: 4. Figures 1 and 3 contain extensive white areas - do these represent missing outage data? If so, this could significantly impact your results as many impoverished counties fall within these areas. Moreover, the presented data doesn't visually support your conclusion that "power outages are more severe in poor counties." Please address this discrepancy.

Response:

Thanks for this insightful comment. The white areas in Figures 1 and 3 represent regions for which outage data were unavailable in our dataset. We have clarified this in the notes accompanying the figures: ‘Areas depicted in white indicate a lack of available power outage data’. And then we primarily supplement our analysis in the following two aspects.

First, despite the missing data in certain regions, our results still reflect the conditions across the majority of China. Specifically, our study utilizes a substantial dataset encompassing 2,245 county-level administrative divisions in China, representing approximately 79% of the national total of 2,843 counties. This extensive sample covers 85.24% of the national GDP and 86.18% of the total population, thereby ensuring that our analysis fully reflects the overall situation in China.

Second, to further address potential concerns arising from missing power outage data, this study employs the Heckman two-step estimation method to test the robustness of the results. Specifically, our collection of power outage data from utility websites may be subject to sample selection bias, as poverty counties with poor digital infrastructure are systematically more likely to fail to disclose outage information online. The Heckman two-step method, proposed by Nobel laureate James Heckman, is a common and effective solution to solve the issue of sample selection bias (Feldman et al., 2015, Heckman, 1976, Wang et al., 2023).

The Heckman two-step method involves the following steps: First, a selection equation is estimated, which is a probability model for whether an individual is included in the sample. This is typically a Probit model that includes all factors influencing the selection process. Second, an outcome equation is estimated, which examines the relationship between the dependent variable and the independent variables. In this step, the inverse Mills ratio (IMR) derived from the first step is incorporated to correct for potential selection bias.

To implement the Heckman two-step model, we require an exclusion restriction variable that affects sample selection but not the outcome variable. This study uses the duration of counties' participation in China's Grassroots Government Information Disclosure Standardization Pilot Program as the exclusion restriction variable. The pilot program, initiated on May 9, 2017, by the General Office of the State Council, designating specific counties as pilot areas(GOSC, 2017), was followed by nationwide implementation on December 26, 2019, to promote the standardization and normalization of grassroots government information disclosure(GOSC, 2019). We code non-participating counties as 0, with values increasing by 1 for each year of participation (1=first year, 2=second year, etc.). Participation in this pilot program significantly increased the probability of power grid-related information disclosure by county governments, as clearly documented in local policy documents. For example, the Notice issued by the Office of the People's Government of Jiangbei District, Ningbo City, on the Issuance of the Implementation Plan for the Comprehensive Promotion of Standardization and Normalization of Grassroots Government Affairs Disclosure in Jiangbei District(JBGO, 2020), as well as the Notification of the Office of the People's Government of Tianhe District, Guangzhou Municipality, on the Issuance of the Implementation Plan for the Comprehensive Promotion of Standardization and Normalization of Grassroots Government Affairs Disclosure in Tianhe District(THGO, 2020), both explicitly stipulate the improvement of information disclosure related to the power grid.

Since the Heckman two-step method does not support high-dimensional fixed effects, we first obtained the power outage residuals from a panel regression that controlled for county-month-by-year fixed effects and other control variables using Models (2) and (4). Then, we applied the Heckman two-step method to these power outage residuals. Model (1) is the selection equation, with the duration of participating in the pilot program serving as the exclusive constraint variable. Models (3) and (5) are outcome equations, with the inverse Mills ratio included for regression analysis.

$$pro\{coll_{it} = 1\} = \alpha_0 + \alpha_1 z_{id} + \alpha_2 Natural\ disaster_{id} + \mu_{id} \quad (1)$$

$$\ln(Outage_frequency_{id}) = X'\theta + \delta_{iym} + \varepsilon_{idf} \quad (2)$$

$$\varepsilon_{idf} = \beta_1 Natural\ disaster_{id} + \gamma_1 IMR_{id} + \varepsilon_{id} \quad (3)$$

$$\ln(Outage_duration_{id}) = X'\theta + \delta_{iym} + \varepsilon_{idd} \quad (4)$$

$$\varepsilon_{idd} = \beta_2 Natural\ disaster_{id} + \gamma_2 IMR_{id} + \varepsilon_{id} \quad (5)$$

Where $pro\{coll_{it} = 1\}$ refers to whether power outage records were collected in this study. The exclusive restriction variable, denoted as z_{id} , represents the duration of participation in the pilot program. X' represents the remaining control variables. IMR_{id} is obtained from the Models (2) and (4), which can effectively adjust the sample selection bias.

We present the results of addressing the sample selection bias using the Heckman two-step model in Table R2. We find that natural disasters have a statistically significant and positive impact on power outages (coef. = 0.0501***; 0.0847***; Columns 1 and 2, Table R2). The results are consistent with our main results, indicating their robustness. We also examine the unequal power outages caused by natural disasters by introducing interaction terms of natural disasters and the dummy variable of poverty counties (Columns 3 and 4, Table R2). The interaction term 'Natural disaster*Poverty' is significant (coef. = 0.0405***; 0.0942***), indicating that the poverty counties experience more outage frequency and longer outage duration caused by natural disasters, showing the robustness of the results.

Table R2. The results of the Heckman two-step method

	(1)	(2)	(3)	(4)
	ln(Outage frequency)	ln(Outage duration)	ln(Outage frequency)	ln(Outage duration)
1st stage (Selection Model)				
Pilot	0.0591*** (0.0016)	0.0587*** (0.0016)	0.0581*** (0.0017)	0.0577*** (0.0017)
2nd stage (Valuation Model)				
Natural disaster	0.0501** (0.0025)	0.0847*** (0.0053)	0.0372*** (0.0021)	0.0560*** (0.0044)
Natural disaster*Poverty			0.0405*** (0.0059)	0.0942*** (0.0124)
Control	YES	YES	YES	YES
County*YM	YES	YES	YES	YES

Notes: * $p < 0.10$, ** $p < 0.05$, *** $p < 0.01$. Standard errors are clustered at the county level and reported below the coefficients. Natural disaster represents the intensity of natural disasters. Poverty is a dummy variable, equaling 1 if the county is a poverty county. Pilot represents the duration of participation in the pilot program. County*YM is the county-month-by-year fixed effect.

Additionally, owing to the multifactorial causes of power outages, the visual description may not visually demonstrate more severe power outages in poor counties. Figure 1 only provides a descriptive overview of the raw spatial distributions of reported power outages and natural disasters across China. The primary focus of this study is to examine the differential marginal impacts of natural disasters on power outage between poverty and non-poverty counties using econometric models. Specifically, we compare how each additional unit increase in disaster intensity affects power outages between poverty and non-poverty counties based on econometric analysis, rather than comparing absolute outage levels using the raw data.

We also provide an intuitive comparison of the absolute levels of average power outage frequency and duration between poverty and non-poverty counties, as shown in the bar chart in the lower-left panel of Figure R1. While it may not be apparent from the maps that poverty counties experience more severe power outages than non-poverty counties, the average data reveals that poverty counties have higher average power outage frequencies and longer average power outage durations.

Editorial Note: The base map used in Figure R1 is the 2019 China map, which has been reviewed and approved with the map review number GS (2019)1822 by the Standard Map Service System of the Ministry of Natural Resources.

Figure R1. Maps of power outages in China

Note: The data is plotted at the county level (2,245 county-level units nationwide, including 457 poverty counties). Panel (a) shows the outage frequency for each county during the study period. The bar chart illustrates the ratio of power outage frequency between poverty and non-poverty counties. Panel (b) shows the power outage duration at the county level. The bar chart illustrates the ratio of power outage duration between poverty and non-poverty counties. Areas depicted in white indicate a lack of available power outage data. The poverty counties in this study were identified by the Chinese government in 2014. The data span from December 2019 to September 2021.

Reference

[1]Feldman ER, Gilson SC, Villalonga B. Do analysts add value when they most can? Evidence from corporate spin-offs. *Strategic Management Journal*. 2015;35:1446-63.

[2]Heckman J. The Common Structure of Statistical Models of Truncation, Sample Selection and Limited Dependent Variables and a Simple Estimator for Such Models. *Annals of Economic and Social Measurement, Volume 5, number 4: National Bureau of Economic Research, Inc; 1976. p. 475-92.*

[3]Wang Z, Lu B, Wang B, Qiu Y, Shi H, Zhang B, et al. Incentive based emergency demand response effectively reduces peak load during heatwave without harm to vulnerable groups. *Nature Communications*. 2023;14:6202.

[4]GOSC. Notification of the General Office of the State Council on the Issuance of the Pilot Work Plan for the Standardization and Normalization of Grassroots Government Affairs Disclosure. In: Council GOotS, editor.2017.

[5]GOSC. Guiding Opinions of the General Office of the State Council on the Comprehensive Promotion of Standardization and Normalization of Grassroots Government Affairs Disclosure. In:

Council GOotS, editor.2019.

[6]JBGO. Notification of the Office of the People's Government of Jiangbei District, Ningbo City, on the Issuance of the Implementation Plan for the Comprehensive Promotion of Standardization and Normalization of Grassroots Government Affairs Disclosure in Jiangbei District. In: Office of the People's Government of Jiangbei District NC, editor.2020.

[7]THGO. Notification of the Office of the People's Government of Tianhe District, Guangzhou Municipality, on the Issuance of the Implementation Plan for the Comprehensive Promotion of Standardization and Normalization of Grassroots Government Affairs Disclosure in Tianhe District. In: Office of the People's Government of Tianhe District GM, editor.2020.

Comments: 5. The presentation format and information included in Table 1 are currently confusing and need revision for better clarity.

Response:

We sincerely appreciate your valuable feedback regarding the clarity and presentation of Table 1. In response, we have undertaken the following revisions to enhance its readability and comprehensibility: (1) Detailed Column Descriptions: We have added comprehensive explanations for each column in Table 1 within the manuscript text to ensure that readers can easily understand the data presented. (2) Consistent Formatting: The table's formatting has been standardized, including uniform alignment and clear labeling of all variables and units, to improve visual clarity and consistency throughout the manuscript. (3) Informative Annotations: We have included detailed footnotes and annotations directly beneath Table 1 to elucidate the definitions of variables and any abbreviations used. Please refer to Pages 5 (lines 28-43) and 6 (lines 1-16) of our revised manuscript for the updated content.

Table 1 presents the regression results. Column (1) shows the impact of natural disasters on power outage frequency. Column (2) shows the impact of natural disasters on power outage duration. Column (3) shows the impact of different types of natural disasters on power outage frequency. Column (4) shows the impact of different types of natural disasters on power outage duration. The regression results indicate that, in all model specifications, natural disasters have a statistically

significant and positive impact on power outages. Specifically, natural disasters increase outage frequency by 4.04% (Column 1, Table 1) and extend outage duration by 5.93% (Column 2, Table 1) on average for an additional unit increase in natural disasters. All models' specifications include county-month-by-year fixed effect (County*YM), which captures the month-by-year-specific characteristics in each county, such as cultural economic conditions, energy transition policies and local infrastructure. We cluster the standard errors at the county level. We further explore the impacts of different types of natural disasters on power outages. We find that strong wind, rainstorms, cold waves, geological hazards, and heatwaves significantly increase outage frequency by 2.98%, 7.99%, 3.77%, 4.45%, and 3.60% (Column 3, Table 1), respectively. Additionally, these natural disasters increase in outage duration by 4.38%, 11.63%, 6.63%, 5.98%, and 4.99% (Column 4, Table 1), respectively.

Table 1. Impacts of grid-related natural disasters on power outages

	(1)	(2)	(3)	(4)
	ln(Outage frequency)	ln(Outage duration)	ln(Outage frequency)	ln(Outage duration)
Natural disaster	0.0404*** (0.0029)	0.0593*** (0.0055)		
Strong wind			0.0298*** (0.0058)	0.0438*** (0.0108)
Rainstorm			0.0799*** (0.0088)	0.1163*** (0.0162)
Cold wave			0.0377*** (0.0046)	0.0663*** (0.0095)
Geo hazard			0.0445*** (0.0103)	0.0598*** (0.0196)
Wildfire			0.0059 (0.0131)	0.0414 (0.0330)
Heatwave			0.0360*** (0.0042)	0.0499*** (0.0082)
Control	YES	YES	YES	YES
County*YM	YES	YES	YES	YES
Observation	1497450	1504150	1497450	1504150
R²	0.8410	0.2505	0.8410	0.2505

Notes: The statistical significance of the coefficients is indicated by asterisks: * denotes $p < 0.10$, ** denotes $p < 0.05$, and *** denotes $p < 0.01$. Standard errors are clustered at the county level and reported below the coefficients. *ln(Outage frequency)* refers to the natural logarithm of power outage frequency. *ln(Outage duration)* refers to the natural logarithm of power outage duration.

Natural disaster represents the intensity of natural disasters, which is the sum of *Strong wind*, *Rainstrom*, *Cold wave*, *Geo hazard*, *Wildfire* and *Heatwave*. *Strong wind* is a dummy variable that equals 1 when the county experiences strong winds; *Rainstrom* is a dummy variable that equals 1 when the county experiences rainstorms; *Cold wave* is a dummy variable that equals 1 when the county experiences cold waves; *Geo hazard* is a dummy variable that equals 1 when the county experiences geological disasters; *Wildfire* is a dummy variable that equals 1 when the county experiences wildfire; *Heatwave* is a dummy variable that equals 1 when the county experiences heatwaves. Controls include dummy variables for holidays and weekends. *County*YM* is the county-month-by-year fixed effect. *Observation* refers to the sample size. R^2 indicates the goodness-of-fit of the regressions.

Comments: 6. Please provide justification for using datasets from different years (2013 poverty data vs. 2020 outage data) and discuss how this temporal discrepancy might affect your analysis.

Response:

We appreciate this critical observation. Under thorough review, we reconfirmed that the classification of poverty counties used in this study is based on the official designation issued by the Chinese government in 2014, known as national poverty counties (SLGPOA, 2014). The 2014 poverty county designation is used in this study for several reasons:

Firstly, there are persistent disparities between poverty and non-poverty counties. While China achieved absolute poverty eradication by 2020, substantial economic disparities between poverty and non-poverty counties persist. As evidenced by the *2020 China County Statistical Yearbook*, non-poverty counties exhibited a per capita GDP (¥62,854) twice that of formerly designated poverty counties (¥31,827). The *Rural Green Book: Analysis and Forecast of China's Rural Economic Situation (2023-2024)* further highlights that the rural disposable incomes in the formerly designated poverty counties still lag behind the national average, standing at only 75.6% of it. Pan et al. (2024) also find that during 2010-2020, despite the poverty-alleviation programs, the gap in public service provision between poverty counties and non-poverty counties still widened.

Secondly, our study aligns with many regional inequality studies in China, using the 2014 classification as the analytical benchmark (Pan et al., 2024, Zhang et al., 2020).

In our study, we utilize the classification standards of poverty counties in 2014 to capture the unequal impacts of natural disasters on power outages in China.

Thirdly, to address the potential temporal mismatch, we also conducted a robustness check of our findings. An alternative variable used to measure regional poverty yielded consistent results. Specifically, we redefined poverty areas using the income in 2020. The poverty area dummy variable was assigned a value of 1 if the income of a region fell within the bottom one-third, and 0 otherwise. The regression results are presented in Table R3. The findings also indicate that poverty counties exhibit a higher frequency and longer duration of power outages caused by natural disasters compared to non-poverty counties.

Table R3 Robustness check based on interaction term

	(1)	(2)
	ln(Outage frequency)	ln(Outage duration)
Natural disaster*Poverty2	0.0172**	0.0300**
	(0.0071)	(0.0142)
Natural disaster	0.0400***	0.0583***
	(0.0038)	(0.0071)
Control	YES	YES
County*YM	YES	YES
Observation	1317220	1321910
R ²	0.8306	0.2481

Notes: * p<0.10, ** p<0.05, *** p<0.01. Standard errors are clustered at the county level and reported below the coefficients. Poverty2 is an alternative definition of poverty counties, valuing 1 if the county falls in the bottom third of income. Control variables include dummy variables for holidays and weekends. County*YM is the county-month-by-year fixed effect. Observation is the sample size. R² represents the goodness-of-fit of the regressions.

We sincerely appreciate the reviewer's thoughtful engagement and meticulous feedback across the revision. Your constructive suggestions have fundamentally strengthened the methodological rigor and interpretive clarity of this work.

Reference

[1]SLGPOA. *The Directory of 832 Nationally Designated Poverty-Stricken Counties (Cities) and Key Counties for Poverty Alleviation and Development in China*. In: *Alleviation SCLGoP*, editor.2014.

[2]Pan Y, Shi K, Zhao Z, Li Y, Wu J. *The effects of China's poverty eradication program on sustainability and inequality*. *Humanities and Social Sciences Communications*. 2024;11:119.

[3]Zhang H, Wu K, Qiu Y, Chan G, Wang S, Zhou D, et al. *Solar photovoltaic interventions have reduced rural poverty in China*. *Nature Communications*. 2020;11:1969.